# Three eruptions at the Fagradalsfjall Volcano in Iceland show rapid and predictable microbial community establishment

Nathan Hadland [1], Christopher W. Hamilton [1], Snædís Björnsdóttir [2] & Solange Duhamel [1,3] ✉

How natural environments transition from uninhabited to inhabited is an incompletely understood question in ecology. We leverage the 2021–2023 Fagradalsfjall eruptions in Iceland as a natural experiment, tracking microbial colonization on the new lava over three years, including lava that had solidified only hours before collection. Samples were collected from fixed sites biweekly for the 2021 eruption phase and then at multiple time points over the next three years, resulting in a unique temporal dataset for primary succession. As the same system erupted again in 2022 and 2023, we were able to monitor a natural ecological triplicate. We use multiple lines of evidence to demonstrate dynamic but predictable community assembly processes. We use alpha and beta diversity, phylogenetic null modeling, taxa volatility, and Bayesian source tracking to propose a two-stage process: (1) rapid establishment of a variable microbial assemblage, followed by (2) stabilization after winter onset. A random forest regression model, trained on 2021 eruption microbial community data, accurately predicts the successional stage in the 2022 and 2023 eruptions. This study underscores the dynamic and predictable nature of microbial colonization in harsh environments, offering insights into primary succession and its role in shaping Earth's ecosystems.

The transition from an uninhabited to an inhabited environment has rarely been documented in nature. Primary succession, or the process of colonizing a bare substrate, is therefore a critical, yet poorly understood ecological process[1–4]. High temporal resolution timelines of microbial community succession have the potential to transform our understanding of ecosystem development and may have important broader applications given that volcanic terrains were potentially the first terrestrial habitats to harbor life on Earth and possibly other Earth-like planets such as Mars[4–8].

Lava is sterile upon emplacement due to its high eruptive temperature (basaltic lava typically erupts at ~1150 °C)[9], and so microbial life is initially absent. Resilient microbial consortia are required for survival in the early stages of colonization due to low nutrient availability[10], extreme temperatures, and low water content[2]. To capture processes shaping microbial community structure, it is important to sample quickly and frequently after the emplacement of a lava flow and conduct periodic fixed-site sampling over time—an approach that has been largely neglected in previous studies and with limited scope (e.g., limited to annual sampling)[11]. Such

longitudinal-type studies enable the investigation of how assembly changes over temporal and biogeographical scales[12]. Prior studies have only sampled lava rocks well after microbial colonization, leaving the transition from an abiotic to a biotic system undocumented[10,11,13–18]. Even the volcanic island of Surtsey in Iceland, formed from 1963 to 1967 during a major submarine eruption and designated as a nature reserve to observe succession, was not comprehensively surveyed for microorganisms until decades after the eruption. Preliminary surveys were conducted shortly after the island's formation and did potentially detect early colonizers such as *Cyanobacteriota* and *Actinomycetota*[19]. By the time broader microbiological investigations were conducted, Surtsey already hosted a diverse microbial community including heterotrophs and members of *Enterobacteriaceae* associated with vegetated areas and bird colonies[20,21]. In addition to monitoring community assembly over time, the mechanisms of microbial deposition remain poorly understood. While some work has attempted to infer the provenance of species identified in volcanic samples[22], rigorous statistical approaches leveraging samples collected from potential source

[1]Lunar and Planetary Laboratory, The University of Arizona, Tucson, AZ, USA. [2]Institute of Life and Environmental Science, University of Iceland, Reykjavik, Iceland. [3]Molecular and Cellular Biology, The University of Arizona, Tucson, AZ, USA. ✉e-mail: duhamel@arizona.edu

environments in the study area (e.g., aerosols, soil, etc.) have yet to be employed[23].

We analyzed the development of microbial communities colonizing freshly emplaced olivine tholeiite basalt lava[24] on the Reykjanes Peninsula in Iceland by conducting fixed-site sampling of lava rocks roughly biweekly for the six-month duration of the eruption, and then each summer for the following three years (i.e., from March 2021 to August 2024). Additionally, the same volcanic system erupted in subsequent years, enabling us to collect lava rocks from three different eruptions in the same environment, resulting in natural replicates for monitoring succession. These separate eruptions will hereafter be referred to as the Fagradalsfjall (active from 19 March to 18 September 2021), Meradalir (active from 3 August to 21 August 2022), and Litli-Hrútur (active from 10 July to 5 August 2023) eruptions. Samples were collected from active lava units soon after solidification and cooling to near-ambient temperature (hereafter "day 1" samples), representing the earliest stage of lava available for microbiological analysis. Near real-time photogrammetric monitoring allowed for high temporal resolution constraints on lava lobe emplacement dates[25], enabling the calculation of lava rock age. We also collected ~2,000–12,500-year-old lava from nearby postglacial lava flows to provide a baseline for what the microbial community in the new lava is headed towards (hereafter referred to as "old lava")[26]. Moreover, samples from potential source environments, including soil, aerosols, and rain were collected near the study site before, during, and after the 2021–2023 eruption period, as well as from hot springs on the Reykjanes peninsula. Combined, this sample suite enabled us to ask two fundamental questions: (1) What are the patterns of microbial community assembly in newly emplaced lava and are those patterns consistent and predictable across multiple eruptions and years? (2) What are the sources of the microorganisms colonizing the newly emplaced lava and how do the communities change over time?

## Results

### Cell abundance increases with lava age

Cell counts by epifluorescence microscopy revealed that cell abundance in the basaltic lava (Supplementary Table 1) increased as a function of lava age (Supplementary Fig. 1A). A linear fit applied to the increase in cell counts resulted in a slope of 182 cells per gram of lava per day over the course of two years ($R^2 = 0.658$; Supplementary Table 2). The maximum measured cell counts from lava samples was $2.2 \times 10^5$ cells/g from Site B (Fig. 1b) at 827 days, the oldest sample measured by microscopy. The minimum was $6.1 \times 10^3$ cells/g from Site 1 at 35 days (Fig. 1a), one of the youngest samples analyzed by microscopy. Cell counts in samples from the Meradalir eruption—which was emplaced against a slope and consisted of lava spatter, which is friable and porous—after one year were comparable to those at the Fagradalsfjall eruption after two years. When plotting cell abundance against environmental variables (pH, water content, lava rock temperature, and weather parameters at the time of sampling; Supplementary Data 1), no correlations were found (all trends had $R^2 < 0.103$; Supplementary Table 2).

In 2022 and 2024, samples were collected from Fagradalsfjall lava proximal to steam vents that formed due to rainwater percolating to the hot core of the lava. The ephemeral fumarole lava had a high rock temperature (~25–30 °C) relative to the ambient. These conditions and access to water from the stream did not correlate with higher cell counts, where abundances remained below the detection limit even after 1 year (Supplementary Fig. 1A). Old lavas were estimated to be around ~$10^9$ cells/gram regardless of the ~10,000-year age difference between old lava flow sites (Supplementary Fig. 1B).

### Young lava contains culturable microorganisms with diverse metabolisms

In total, we isolated 9 strains from young lava and 2 strains from soil on organic media, while none were cultured from old lava (Supplementary Table 3). The lava-derived strains were collected from the Fagradalsfjall eruption at various locations and time points, both in nutrient-poor broth (R2A) and nutrient-rich (tryptic soy broth; TSB)

broth and both at 11 °C (average mid-day ambient temperature at Reykjanes in late Spring) and 50 °C (to select for thermophiles). The youngest sample with successful isolation (56 days) was cultured in nutrient-poor R2A both at 11 °C and at 50 °C and was identified as *Sphingomonas echinoides*, a strictly aerobic, metabolically versatile widespread soil bacterium[27]. Two isolates from both TSB and R2A at 50 °C were identified as *Bacillus licheniformis*, a gram-positive, endospore-forming, mesophilic, facultatively anaerobic strain known for its tolerance to salinity, high temperature, and pressure stress. It is found in soil environments and on bird feathers, including in Antarctica[28]. This species was also isolated from a soil sample. *Brevibacillus borstelensis* was isolated from an 81-day-old sample in R2A at 50 °C and is a known endospore forming thermophile[29]. *Paenibacillus* sp. was isolated from a 94-day-old sample in TSB at 11 °C and is known to have important functions in the rhizosphere[30]. Additional strains were isolated from BG-11 (to isolate phototrophs), sulfur-oxidizer media, and R2A from lava collected from 1-year-old samples, but we were unable to revive them from glycerol stocks and therefore unable to extract enough DNA for identification. Nevertheless, the successful culturing in these media illustrates the presence of a culturable community within the lava exhibiting various metabolic functions. Additionally, subsamples of the lava were subjected to a carbon substrate utilization assay. Elementary sugars and polymers were preferentially consumed by organisms in young lava (Supplementary Fig. 2).

### Community Composition Shifts are Consistent Between Eruptions

Using 16S rRNA gene sequencing, the phyla *Pseudomonadota* and *Actinomycetota* dominated taxonomic diversity in the lava rock samples throughout the study period. *Bacillota*, *Bacteroidota*, *Acidobacteriota*, *Chloroflexota*, and *Eremiobacterota* also had high relative abundances at different lava ages (Fig. 2). Although our study was focused on prokaryotes, there is some overlap in the 16S rRNA gene with eukaryotic sequences. For example, before filtering, mitochondrial and chloroplast sequences were detected in weeks- to years-old samples. The chloroplast sequences appeared in samples as young as day 1. This pattern was consistent across the eruptions. However, these eukaryotic sequences were removed before subsequent analyses.

**Days to months.** Amplicon sequence variants (ASVs) of genera known to include extremophiles were detected in day 1 to several weeks-old samples. *Udaeobacter* is a soil bacterium capable of surviving in oligotrophic conditions[31] that was found in relatively high proportions in young samples at Fagradalsfjall, Meradalir, and Litli-Hrútur including day 1 samples. However, it dropped off after the first winter (Fig. 3a) and was not detected in 1-year or 2-year-old samples. A similar pattern occurred with several other groups. For example, 29% of a day 1 sample was identified as *Paracoccus*, a diverse genus known to contain chemolithoautotrophs[32], but it dropped off within weeks. Weeks- to months-old community composition was variable. For example, a 45-day-old sample from Site 2 (Fig. 1a) did not have a single taxon with relative abundance above 4.1%. The only archaeal ASV was found in samples less than 100 days old (up to 7.8% relative abundance), and identified as the genus *Nitrososphaera*, an ammonia-oxidizing mesophile found in soils, hot springs, and freshwater and marine sediments[33].

We established multiple sites that were sourced from the same eruptive vent but were spatially separated to ensure sites were not overprinted by subsequent lava flows. Consequently, the site location was not part of the initial experimental design and therefore can be considered as a random effect for the purposes of statistical analysis[34]. To elucidate trends in taxa abundance without this influence of spatial variability, linear mixed effects (LME) models were applied to certain taxa. Using this model for *Udaeobacter* (Fig. 3a) on samples <100 days, a statistically significant ($n = 35$;

**Fig. 1 | Overview of the study area. a** Fagradalsfjall (blue-gray), 2022 Meradalir (orange), and 2023 Litli-Hrútur (green) eruptions with a background hillshade derived from previous work[25,69,70]. A total of 18 Fagradalsfjall, 3 Meradalir, 4 Litli-Hrútur sampling sites were established, though some are too closely spaced to be individually resolved on the map. Note that sampling sites 1, 2, 3, and 4 were from Fagradalsfjall, but were later resurfaced by new lava flows from Meradalir. Earlier in the eruption, day 1 sites were established near the eruptive vent but quickly covered. Additional sites were collected in 2024 to determine spatial variability, indicated by the black circles for Fagradalsfjall, pink triangles for Meradalir, and light green × for Litli-Hrútur. **b** Inset of area around Sites A–C. A late-stage breakout in September of 2021 covered Site A and the area of the flow is shown in transparent white.

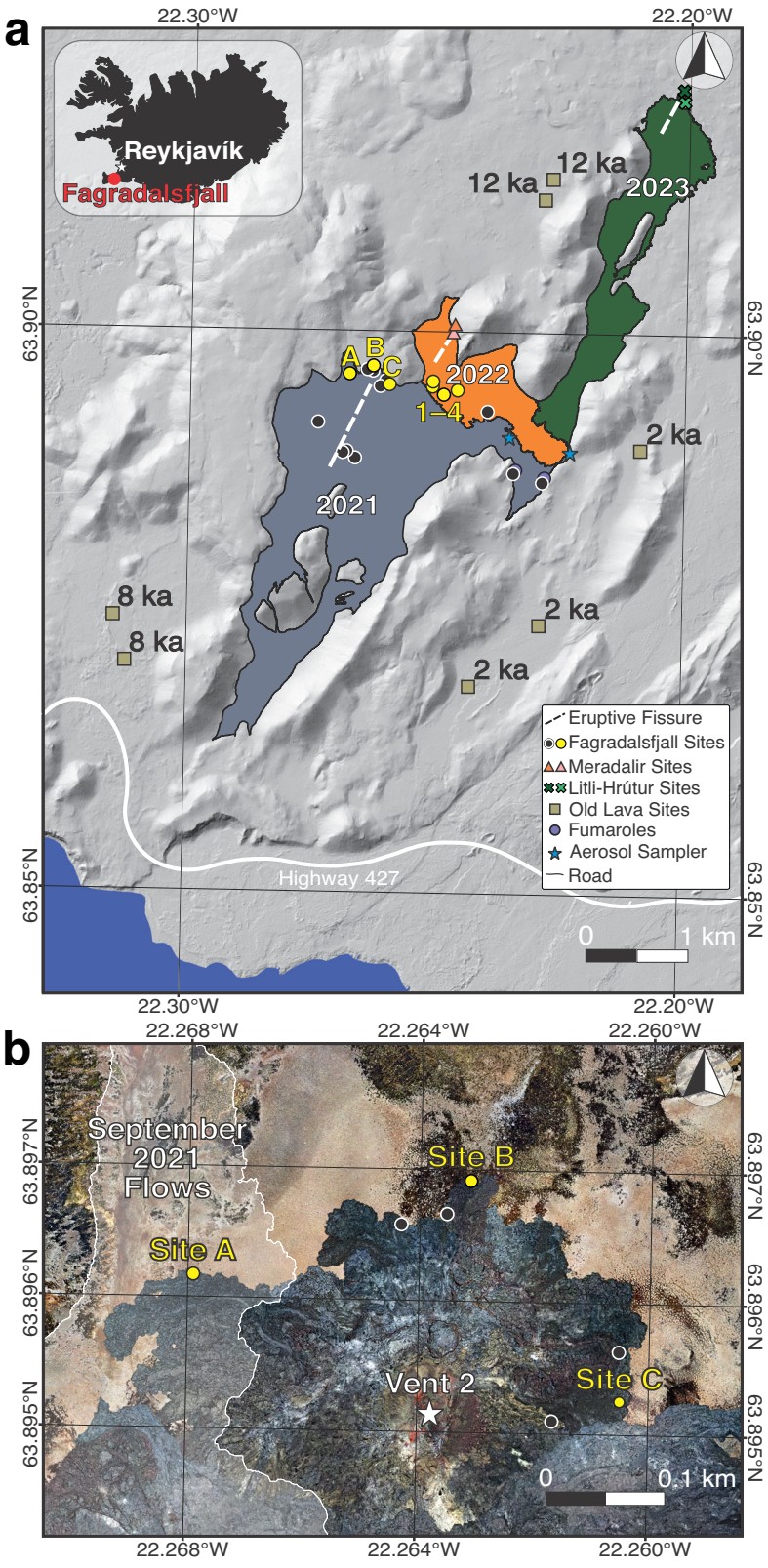

$p = 0.031$) increase was observed. The site location accounted for 40.8% of the variance in the data (Supplementary Table 4).

**1 to 3 years**. In the 1- and 2-year-old samples, there were major shifts compared to samples <100 days old. Since all the 1- to 3-year-old samples were collected in the summer (temperature in 2021–2024 was 9.97 ± 2.25 °C,

mean ± s.d.), these shifts occurred after the winter (average temperature was 0.10 ± 3.85 °C). Moreover, these shifts were consistent between the three eruption sites. For example, *Micrococcales* saw a steady decline over the course of the study period (Fig. 3b). This order is a diverse group that has been documented in successional processes in glacial forefields (i.e., secondary succession)[35,36], found in dust plumes[37], and

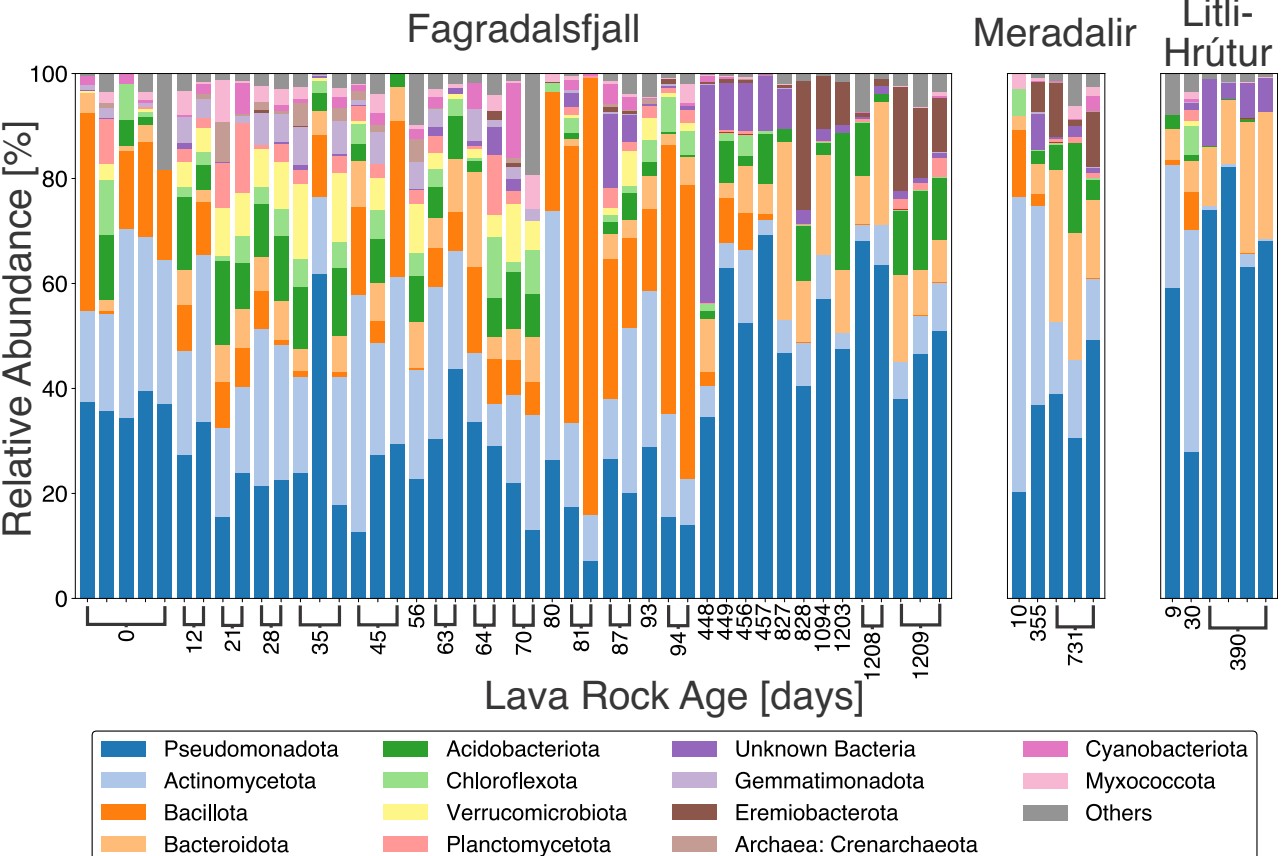

**Fig. 2 | Microbial community composition of the lava as a function of age.** Variation in prokaryote phyla relative abundance as a function of lava rock age and separated by eruption. Less abundant phyla were concatenated under "Others" but can be found in detail in the Supplementary Data 2.

cultured from atmospheric and volcanic samples in Iceland[38], though some members have been suggested to be potential human commensals[39]. While it had high relative abundances in all three lava flows, a drop-off was observed after one year and it was not detected in years 2 and 3. An LME model applied over the course of the entire 3-year study period found a statistically significant ($n = 58$; $p < 0.001$) negative correlation between lava rock age and the relative abundance of the *Micrococcales*. Conversely, statistically significant increases were found for the genus *Acidiphilium*, which is typically found in highly acidic environments[40], and for the methanotrophic family *Beijerinckiaceae* (Fig. 3c, d and Supplementary Table 4)[41]. *Eremiobacterota* also increased in proportion after the first winter. This phylum exhibits acidotolerance and chemolithotrophy, and has been found on Mt. Erebus in Antarctica and involved in trace gas oxidation and anoxygenic photosynthesis[42,43]. Despite the high proportions of putative acidotolerant organisms in the 2- and 3-year-old samples, the bulk pH of the lava was circumneutral (Supplementary Data 1), though localized acidic conditions in the pore spaces are possible.

**Fumaroles.** Fagradalsfjall lava proximal to ephemeral steam vents that formed due to rainwater had a few dominating ASVs that were comparable to the few days-old samples (e.g., *Micrococcales*), despite being 1 year old. Once the system had cooled to ambient temperatures by 2024, groups such as *Eremiobacterota* and *Acidiphilium* became prominent (Supplementary Fig. 3).

**Alpha and beta diversity shifts are consistent between eruptions**
Alpha diversity was variable in samples <100 days old based on Faith's phylogenetic diversity (PD). The lowest Faith's PD values were observed in the day 1 samples. A sample collected from lava that was still hot inside a

crack (139 °C) but the ambient temperature on the surface had the lowest Faith's PD value, where we were still able to successfully amplify DNA and recover non-contaminant sequences. Faith's PD also varied considerably within sites during the first 100 days. After the first winter, the diversity values at Sites B and C became more consistent (Supplementary Fig. 4A). To elucidate statistical trends, an LME model was applied to Faith's PD in samples <100 days, treating the site as a random effect. The model revealed a statistically significant increase in Faith's PD with lava rock age ($p = 0.031$). In the LME model, the site accounted for ~40.8% of the variance in the data. When including the old lava, Faith's PD significantly increased with lava age over a timescale of thousands of years ($n = 70$; Spearman's rank test correlation coefficient = 0.3722, $p = 0.002$).

Ecological patterns were further explored using beta diversity analysis, specifically principal coordinate analysis (PCoA) using the distance matrix of the weighted Unifrac metric, which considers the presence/absence and relative abundance of taxa, along with their phylogenetic relationships. The lava samples grouped as a function of lava rock age. Samples <100 days old clustered together while 1-year-old and 2- to 3-year-old samples were separated (Fig. 4a). To understand how the clustering changed as a function of lava rock age while accounting for the sample site, the distance between successive samplings was calculated (Fig. 3e). The Unifrac distance to the previous sampling for individual sites showed moderate shifts in the first 100 days, followed by a large shift between these samples and those collected after 1-year following the first winter. However, there was only a moderate shift between the 1- and 2-year-old samples, and a smaller shift between the 2- and 3-year-old samples. Next, we used the Jaccard metric which measures the proportion of shared elements, considering only the presence or absence of taxa (Fig. 4b). The Jaccard distance between successive time points (Fig. 3f) shows that many species at one time point were absent at the successive

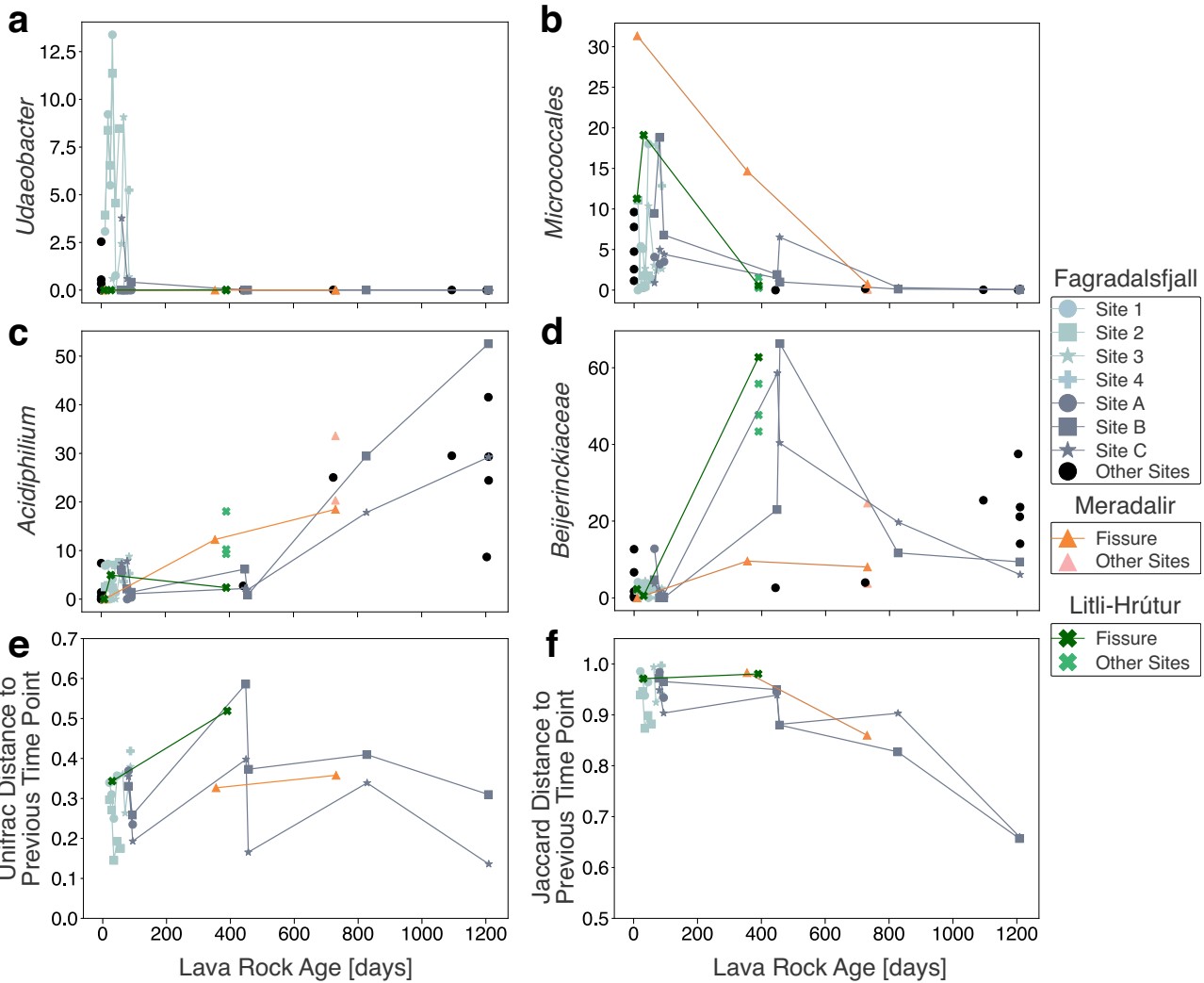

**Fig. 3 | Longitudinal volatility analysis of different metrics of the young lava rock samples as a function of lava rock age.** Relative abundance (%) of specific features including (**a**) the genus Udaeobacter, (**b**) the order Micrococcales, (**c**) the genus Acidiphilium, and (**d**) the family Beijerinckiaceae. Volatility analysis was also applied to beta diversity metrics including (**e**) weighted Unifrac and (**f**) Jaccard by time point in the younger ages. However, the distance decreases over time, indicating an increase in shared taxa in successive years.

measuring the distance to the previous time point at the same fixed site. Lower distances equate to higher similarity to the previous time point. Sites at Fagradalsfjall are colored according to spatial proximity whereas Meradalir and Litli-Hrútur have distinct symbols and colors.

**Sources of microorganisms to the new lava flows**

Samples of soil, bioaerosol, hot spring, rain, and old lava (i.e., potential sources of microorganisms) were collected throughout the course of the eruption and in successive years to determine the origins of the microbial community in the new lava. The algorithm SourceTracker2 implements a Bayesian inference model to iteratively calculate the probability that a sequence in a sink (e.g., the young lava) originated from specified sources[23]. This model requires that the defined sources have distinct microbial fingerprints[44]. These associations are discussed in the Supplementary Material (Supplementary Figs. 5, 6), and we proceeded with the broad categorical classifications of the sources.

SourceTracker2 analysis revealed that the initial colonizers of the lava were sourced from a combination of bioaerosols and soil particles (Fig. 5). However, large proportions were attributed to an unknown source. Following the first winter, there was a major shift in the sources of microorganisms, paralleling trends observed in taxa relative abundance, alpha diversity, and beta diversity. After one year, rain emerged as the primary source of microbial colonization, contributing up to 98 ± 0.003% (mean ±

s.d.) of the inferred source environments. These shifts were consistent across all three eruptions, though Litli-Hrútur had a higher proportion of soil-derived microorganisms in the 1-year-old samples in addition to rain, up to 22 ± 0.04%. Old lava was only a minor source, and the hot springs (collected from nearby Krýsuvík Geothermal Area) did not source microorganisms. Qualitative assessments of source and sink similarities using the weighted Unifrac metric showed that the old lava, soil, and aerosols generally clustered with the youngest lava samples <100 days old (Supplementary Fig. 6). In contrast, the 1- to 3-year-old samples clustered closely with rain. Together, these associations confirm the SourceTracker2 analysis.

**Processes of microbial community assembly**

The integration of phylogenetic analyses with null models has facilitated the inference of niche and neutral processes (e.g., deterministic vs. stochastic) in microbial community assembly[45,46]. Inferring these ecological mechanisms requires determining the phylogenetic signal. This means examining whether the ecological similarity among taxa is related to their clustering on a phylogenetic tree[47]. Mantel correlograms, which correlate phylogenetic distance with niche distances, detected significantly positive phylogenetic signals for lava rock age ($p < 0.05$; Supplementary Fig. 7), thereby supporting this assumption. We used age as a proxy for cumulative environmental

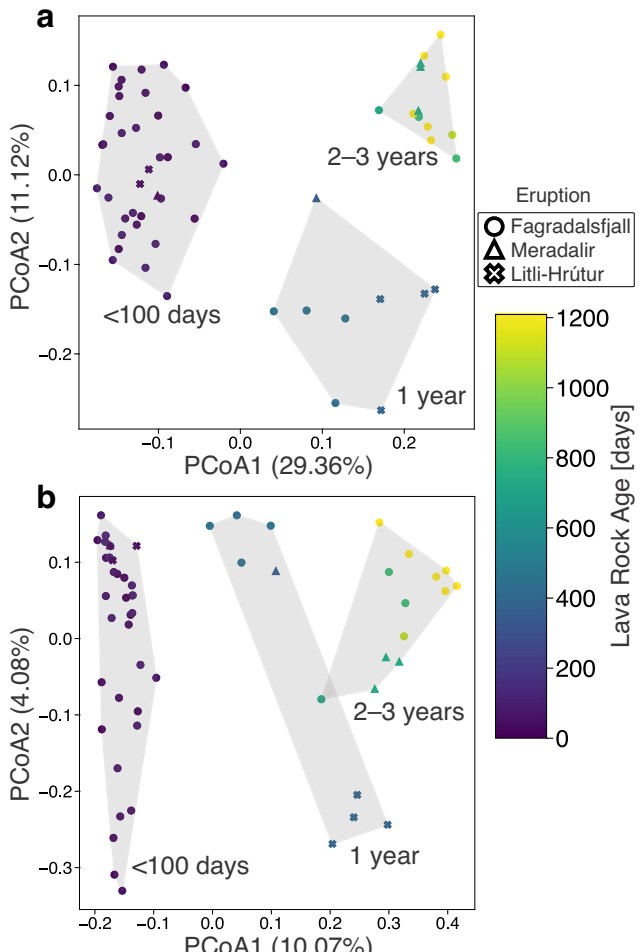

**Fig. 4 | Principal coordinate analyses (PCoA) of beta diversity. a** Weighted Unifrac and (**b**) Jaccard metrics capturing 40.48% and 14.15% of the variation in multivariate space, respectively. The variation explained for the individual principal coordinates are listed on the respective axis legends. Convex hulls, which enclose the outermost points of a group to define its boundary in multivariate space, were drawn around samples grouped by age bins (<100 days, 1 year, and 2–3 years old) to highlight patterns in community composition across different successional stages. These groupings illustrate the clustering of microbial communities as they develop over time.

and stochastic processes by comparing communities in the same age bin. Because of the limited number of samples from Meradalir and Litli-Hrútur for pairwise comparisons, we focused on Fagradalsfjall. We identified two distinct periods (Fig. 6c). The first 100 days were dominated by homogenous selection (constant environmental factors lead to low turnover in the community) and drift (fluctuations in population due to random events). There was also some minor contribution of homogenous dispersal (e.g., dispersal was sufficient to overshadow deterministic processes) in the day 1 bin. After the first winter, the assembly processes shifted towards dispersal limitation, a stochastic process limiting organism exchange between communities, resulting in divergence in community compositions due to drift. Shifts in assembly processes after the first winter were also observed using βNTI (Supplementary Fig. 8A).

**Prediction of lava rock age using a random forest regressor**
Random forest regressors are employed to predict continuous metadata values by learning trends in microbial community composition from a labeled training set and applying the model to a test set. We used the model on microbiome data from the Fagradalsfjall eruption at the order taxonomic level (training set; $n = 46$) to predict the ages of lava samples from the Meradalir and Litli-Hrútur eruptions (test set; $n = 11$). The model demonstrated a robust predictive performance ($R^2 = 0.95$), indicating a very strong positive linear relationship between the predicted and true ages (Fig. 7a). The model's prediction errors were quantified using the mean absolute error (MAE) and root mean squared error (RMSE), which were 89.6 days and 96.3 days, respectively. Additionally, a paired $t$ test revealed a statistically significant systematic overestimation of the predicted ages compared to the true ages ($t$-statistic = 3.634, $p = 0.005$). This indicates that the model tended to predict slightly higher ages than the actual values. Despite this bias, the strong correlation and high $R^2$ value demonstrate that the model effectively captured the patterns of microbial community development across consecutive years.

To explore whether inferred metabolic trait shifts could predict lava rock age, predicted pathway abundances for each sample using the paprica[50], PICRUSt2[51], and FAPROTAX[52] tools were used (Fig. 7b), as done previously for predicting biogeochemical processes[53] to provide a complementary view of community potential. These models had a lower performance than the absolute abundance data (paprica: $R^2 = 0.64$; PICRUSt2: $R^2 = 0.69$; FAPROTAX: $R^2 = 0.79$) and had an overestimation bias (paprica: MAE = 188.8, RMSE = 225.2; PICRUSt2: MAE = 136.0, RMSE = 171.9; FAPRO-TAX: MAE = 314.8, RMSE = 388.1). While metabolic inference tools must be treated with caution due to their reliance on curated databases, cultured isolate characterization, or reference genomes[54,55], the inclusion of inferred metabolisms nonetheless serves as a valuable exploratory approach and foundation for hypothesis generation regarding functional shifts over time. As an additional example, PICRUSt2-estimated 16S rRNA gene copy number decreased over time across all eruptions (LME model, $p < 0.001$; Supplementary Fig. 9), indicating a shift toward slower-growing micro-organisms, consistent with patterns observed in successional studies such as Nemergut et al.[56]. However, the overall copy numbers remained comparatively low.

**Discussion**
Multiple metrics revealed that the lava flows analyzed in this study rapidly hosted microorganisms within hours and days of solidification. Cell counts indicated that the Fagradalsfjall lavas collected in 2021 (<100 days old) represent one of the lowest biomass environments reported on the surface of Earth (Supplementary Fig. 1A), comparable to the hyperarid core of the Atacama Desert[57]. It is important to note that cells were extracted from powdered bulk rock, which likely resulted in lower cell abundance per gram of lava rock, compared to the surface of the rock, which is expected to be colonized first[58]. Moreover, since our data cannot distinguish microorganisms growing on the lava (i.e. metabolizing lava rock resources), it remains unclear whether microorganisms were established or merely transient on the surface of the lava, including dead cells. For example, Sites 1–4 were

effects because other physicochemical parameters (e.g., pH, mineralogy) remained relatively constant across sites (Supplementary Table 1 and Supplementary Data 1).

Under the assumption of a phylogenetic signal, ecological processes governing the phylogenetic structure of individual lava samples were investigated using the nearest taxon index (NTI) and the net relatedness index (NRI). Similar to trends observed in Faith's PD, both the NTI and NRI showed high variability in samples less than 100 days old (Fig. 6a, b). The mostly positive values indicate a dominance of phylogenetic clustering[48]. An LME model applied to the first 100 days found no correlation with lava rock age, but there was a statistically significant decrease over the entire study period for both NTI and NRI ($p < 0.001$; Supplementary Table 4). A few samples had NRI values less than zero, indicating competitive exclusion; but most values clustered around zero or were weakly positive, indicating that the phylogenetic structure approached the null expectation and stochasticity dominated after the first winter. Conversely, although NTI values decreased, they remained positive.

The beta diversity component of NRI (βNRI), which is the phylogenetic distance between samples compared to the null distribution[46,48], and the Raup Crick (RC) diversity metric[49] were used to partition deterministic

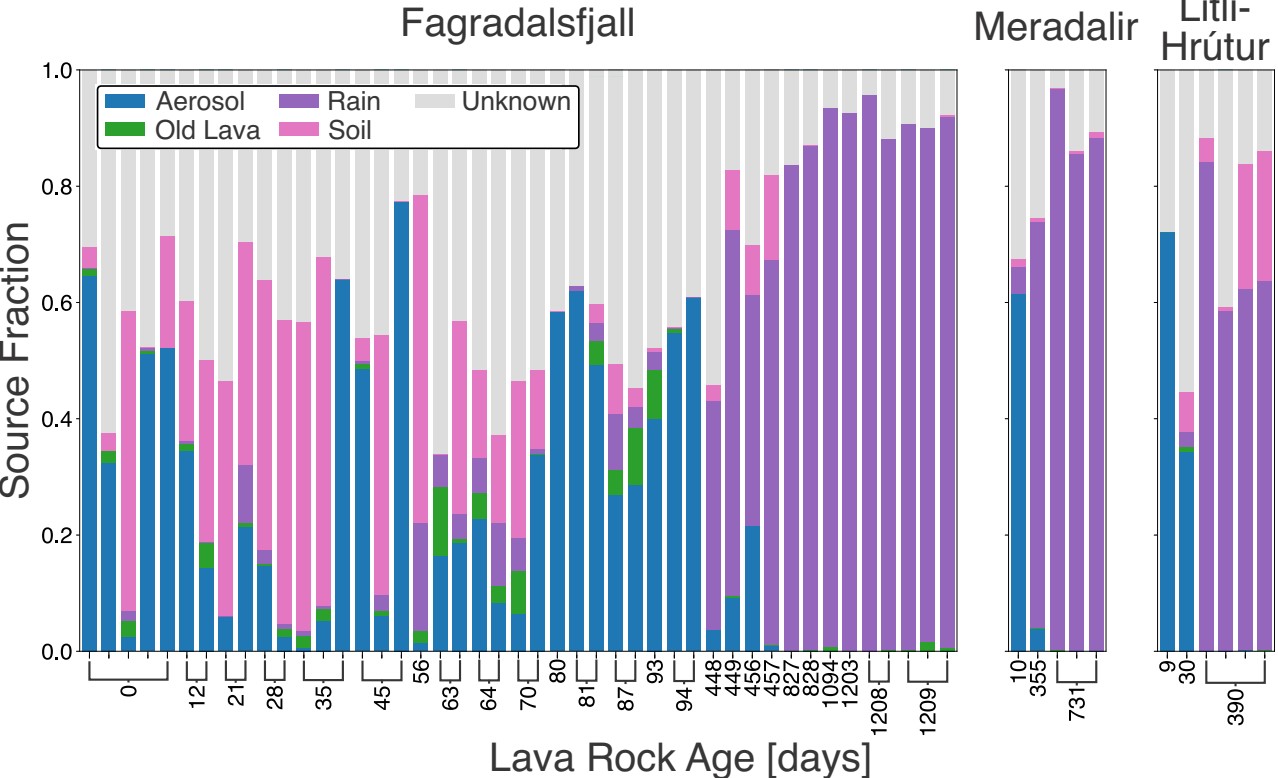

**Fig. 5 | Quantitative source tracking of microorganisms colonizing the lava using a Bayesian model.** Plots are separated according to eruption. Each color represents the mean fractional contributions of each source to each sink (lava sample). While hot springs were included as a source in the model, they were found to not source microorganisms and therefore were not included.

emplaced against a steep hyaloclastite slope and within confined topography (Fig. 1a). Mass wasting may have deposited soil particles onto the surface of the lava flow, which would explain the higher proportion of soil in the SourceTracker2 results (Fig. 5). Similarly, bioaerosols could have been transiently present on the surface of the rocks. However, the consistent presence of certain taxa from the very first sample and throughout subsequent ones (e.g., *Udaeobacter*; Fig. 3a and *Paracoccus*) illustrates the potential colonization of freshly solidified lava by microorganisms adapted to survive harsh conditions. The successful culturing of microorganisms from the young lava and heterotrophic activity measured in the substrate utilization assay also demonstrate the presence of viable microorganisms.

Dispersal to new habitats imposes significant environmental stressors on microorganisms. The youngest samples exhibited high NTI and NRI values (Fig. 6a, b). This indicates that phylogenetic clustering was prominent early on and that environmental conditions initially selected for species capable of surviving and persisting in these conditions[45,47]. Similarly, the dominance of homogenous selection in these early stages (Fig. 6c) indicates strong environmental filtering consistent across sites. The presence of drift, particularly in the day 1 bin, implies that stochastic events, such as the random arrival of certain microbial species, played a role. Therefore, the first microbial colonizers were likely those that fortuitously arrived through dispersal events (e.g., wind) and were well adapted to a variety of extreme environmental conditions. Microbial bioaerosols possess adaptations such as cell pigmentation, endospore formation, tolerance to oligotrophic (nutrient-poor) conditions, and protection against desiccation and solar radiation[59]. These selection pressures are also faced by microorganisms in volcanic terrains[4], which may explain why some of the culturable isolates from the lava samples were determined to be endospore formers that could grow on nutrient-poor broth (Supplementary Table 3). Combined with the dominance of bioaerosol-contributed microorganisms in samples <100 days, as indicated by the SourceTracker2 results (Fig. 5), these findings

illustrate the initial stage in community assembly in this system. That is colonization by microorganisms well suited to oligotrophic and harsh environmental conditions, along with an increase in phylogenetic diversity (as reflected in Faith's PD) as additional species were deposited. The warmer summer conditions on the Reykjanes Peninsula may have further contributed to the increase in Faith's PD.

Following the first winter, a dramatic shift was observed in all metrics analyzed. In the longitudinal analyses of beta diversity (Fig. 3e, f), the distance to the previous time point was notably high after 1 year in both weighted Unifrac and Jaccard metrics because of an increase in phylogenetic/feature dissimilarity, which suggests turnover in the community. Moreover, the clear clustering based on sampling year (Fig. 4), regardless of the eruption, points to consistent shifts in community composition driven by seasonal shifts. In years 2 and 3, however, samples clustered together and the Jaccard distance decreased, suggesting a degree of stabilization in the community during these subsequent years. This stabilization is further evidenced by a lower but more consistent Faith's PD (Supplementary Fig. 4). Certain taxa such as the methanotrophic family *Beijerinckiaceae*[41] or the acidotolerant and potentially iron-reducing genus *Acidiphilium*[40] became dominant in these subsequent years (Fig. 3c, d), while others like *Micrococcales* (Fig. 3b) steadily declined. These taxa exhibited the same trends in the Meradalir and Litli-Hrútur eruptions, indicating that similar successional patterns were occurring across different eruptions. This consistency suggests that microbial community development on new lava flows follows a predictable trajectory, influenced by important taxa and seasonal changes. Many of these taxa have been observed in other volcanic systems, such as *Micrococcales* and *Paracoccus* at Fimmvörðuháls in Iceland[38] and *Eremiobacterota* in Hawai'i[18]. Several ASVs could not confidently be assigned to a named phylum, particularly after 1 year in all three eruptions (Fig. 2), akin to microbial dark matter in lava caves[18]. Future research should focus on using a multi-omics approach to elucidate metabolic relationships among these

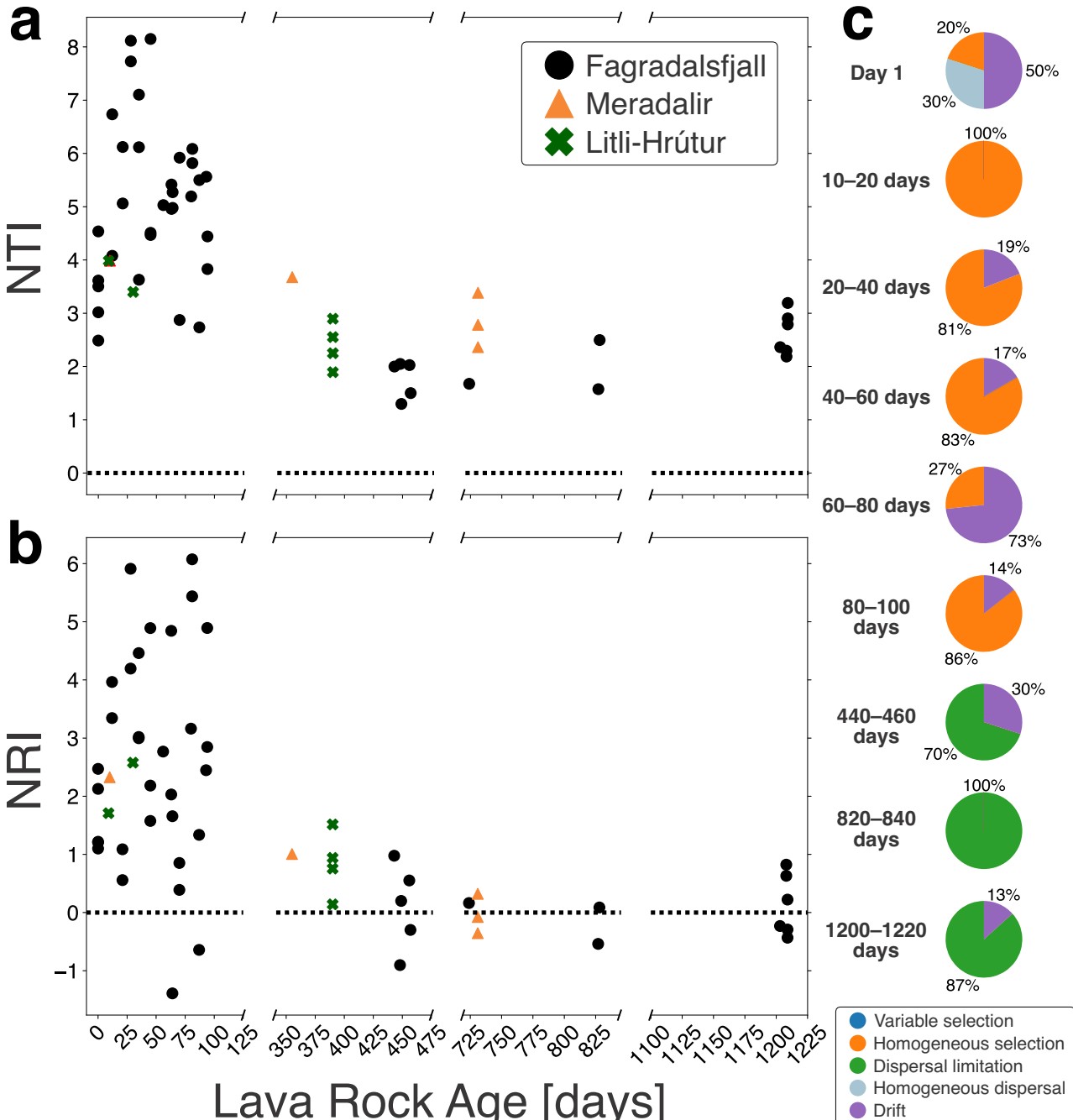

**Fig. 6 | Phylogenetic null modeling results. a** The nearest taxon index (NTI) and (**b**) the net relatedness index (NRI) as functions of lava rock age (in days). The *x*-axis is presented on a broken scale to highlight changes over different time intervals. The horizontal dashed line at zero denotes the transition from phylogenetic clustering (NRI or NTI > 0), associated with strong environmental filtering, to phylogenetic overdispersion (NRI or NTI < 0), associated with competitive exclusion. **c** Processes of microbial community assembly for Fagradalsfjall samples at different age bins using the beta diversity component of NRI ($\beta$NRI) to partition deterministic processes where $|\beta$NRI$| > 2$, and the Raup Crick (RC) diversity metric to partition stochastic processes where $|\beta$NRI$| < 2$.

taxa, which could lend further insight into community structure, functioning, and assembly processes.

These shifts in taxa abundance after the first winter did not occur in the Fagradalsfjall lava proximal to ephemeral fumaroles. While the fumaroles formed by meteoritic water provided moisture and warmth, they exhibited low cell counts and a community composition similar to day 1 samples despite being 1 year old (Supplementary Fig. 3). The slightly elevated temperatures may have been too prohibitive for more diverse colonization until it cooled to ambient temperature (Supplementary Figs. 1, 4, 6). Ephemeral fumaroles have been suggested to provide abodes of habitability

in extreme systems, particularly with relevance to Mars which was volcanically active in the past[4,8]. However, akin to studies on fumaroles in temperate environments, the surrounding lava may have imposed fewer selection pressures, resulting in comparatively higher biomass and diversity[3,60]. Consequently, localized environmental stress might have suppressed succession, and the elevated temperatures may have restricted colonization to a narrow set of stress-tolerant taxa with limited niche width, slowing community turnover.

Source tracking (Fig. 5) revealed that, after the first winter, rainwater became the predominant contributor to microbial assemblages on the lava

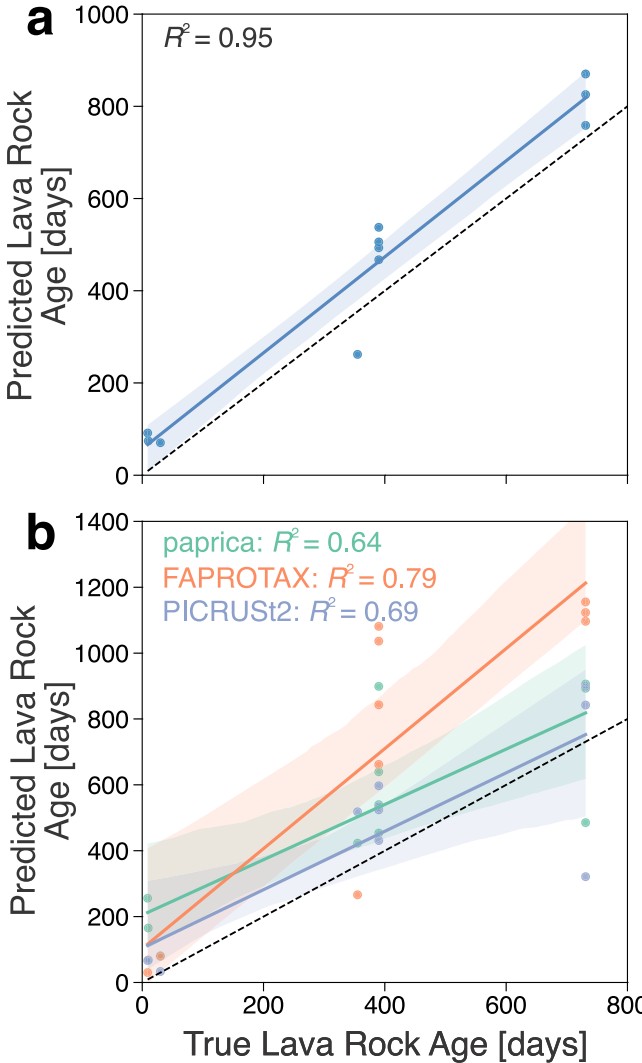

**Fig. 7 | Random forest regressor scatter plots.** Predicted vs. true values for each test sample (Meradalir and Litli-Hrútur eruptions) using (**a**) order-level taxonomic features and (**b**) inferred metabolisms with the paprica[50], FAPROTAX[52], and PICRUSt2[51] pipelines. The plot is accompanied by a linear regression fitted to the data with a 95% confidence interval (shading). The true 1:1 ratio between predicted lava rock age and true values is represented by a dotted line.

flows across all three eruptions, highlighting the increasing influence of precipitation on microbial community composition. This transition potentially reflects a broader ecological shift toward more stable microbial establishment and the emergence of biotic interactions. Notably, these findings underscore the role of rain as a vector for microbial dispersal. The influx of microorganisms via rain coincided with decreasing NTI and NRI values, bringing phylogenetic distances closer to the null expectation (Fig. 6a, b)[45,47,48] and indicating a weakening of strong environmental filters. Concurrently, the shift towards dispersal limitation (Fig. 6c) indicates that the establishment of new species was increasingly governed by the presence of local species and their ability to colonize the lava surfaces. Previous work similarly found that dispersal limitation strongly shapes microbial community structure in Antarctica[61]. Together, these findings point to the onset of the second stage in the community assembly—a stabilization phase where locally sourced species established unique niches within an increasingly mature ecosystem. The community shifted from being shaped primarily by strong environmental filters and stochastic colonization to one increasingly influenced by species interactions, competition, and niche specialization. We similarly observed consistent Faith's PD and inferred cell abundance values in old lava despite thousand-year age differences (Supplementary

Fig. 1B, 4B), further supporting the stabilization trend over longer timescales.

LME models applied to a variety of metrics found that the site location accounted for some of the variance in the data (e.g., 40.8% for Faith's PD; Supplementary Table 4), underscoring some influence of local environmental conditions. Some of this site-to-site variability may reflect differences in lava morphology and microenvironmental exposure[4]. For instance, porous spatter from the Meradalir eruption appeared to support more rapid colonization than Fagradalsfjall samples, with elevated cell counts after one year, potentially due to increased surface area, friability, or slope-driven soil input (Supplementary Fig. 1A). Additionally, while all three eruptions potentially originated from similar mantle regions[62] and our samples exhibited broadly similar mineralogy (Supplementary Table 1), subtle geochemical differences—possibly reflecting variable source contributions[24,63]—could have had compounding effects in microbial community assembly[64].

Despite these possible spatial heterogeneities, lava age and seasonal shifts still selected for important taxa and the community had consistent structural shifts over time regardless of the site or eruption. Our random forest model, trained on microbiome data from the Fagradalsfjall eruption, harnessed these trends to predict the stages of microbial community assembly of the Meradalir and Litli-Hrútur eruptions (Fig. 7a). Despite a slight overestimation bias, the model's robust performance ($R^2 = 0.95$) is important for advancing our understanding of microbial ecology. Firstly, the strong predictive performance of the model highlights the potential for ensemble learning approaches to unravel complex ecological patterns and processes that may not be readily apparent through traditional analyses. While there is precedent for predicting timepoints with random forest models, this is typically done in human microbiome studies[65] rather than in the natural environment. Secondly, this data demonstrates that microbial community composition can serve as a reliable indicator of the successional stage, similar to previous work predicting geochemical properties using a random forest[53]. Given adequate training data, we anticipate that this approach could be generalized to predict successional stages in a variety of environments.

While early colonization is likely governed primarily by deterministic processes, the influence of priority effects, biotic interactions, and microenvironmental heterogeneity may compound over longer timescales, potentially limiting the generalizability of machine-learning models trained on early-stage data. For example, Selensky et al.[66] found that despite organic matter inputs, biofilms within lava tubes at Lava Beds National Monument were supported primarily by active carbon fixation. Chemoautotrophs likely were the initial colonizers in the newly formed lava tube and as overlying vegetation was established and organic detritus inputs increased, the original chemoautotroph colonizers may have maintained their independent niche. Similarly, we show that NRI decreased over time and there was a shift towards stochastic assembly (Fig. 6), reflecting the increasing importance of biotic factors (e.g., species interactions) and likely the early stages of compounding priority effects. In decades-old lava, previous work found that differences of a few years did not significantly impact the microbial community, likely due to environmental differences[17]. These findings suggest that spatial heterogeneity, stochastic processes, and priority effects may increase in influence over time—but have not yet overridden the strong temporal signal observed within the three-year window examined here (Fig. 7).

These dynamics also have implications for volcanic systems on Mars, which have been active in the geologically recent past[67]. Understanding the dynamics of rock-hosted ecosystems in analog environments was identified top science priority in the Planetary Science and Astrobiology Decadal Survey 2023–2032[68]. Therefore, this work has implications for both planetary exploration and Earth's own dynamic ecosystems, including early Earth.

In summary, we have characterized how a microbial community developed from a sterile substrate over short temporal scales from hours to years. A two-step model is proposed for the community development of the

lava rocks: (1) Initial random colonization via aerosol and soil by organisms well suited to oligotrophic and harsh environmental conditions; and (2) Subsequent stabilization after the first winter as different species establish unique niches in an increasingly stable and homogenous environment. While slight differences were observed between spatially proximal sites and eruptions, suggesting that micro-environmental parameters also influenced microbial community assembly, these variations fit within the overarching predictable framework of the two-step model. These stages were observed using multiple metrics across three different eruptions occurring in sequential years, effectively representing an ecological triplicate in the natural environment. We were able to harness these consistent trends using a random forest regressor to predict the age of the 2022 and 2023 lava flows, demonstrating that microbial community composition is a reliable indicator of successional stages. These results have important implications for understanding soil development and colonization processes under extreme conditions such as those imposed by lava flows. In terms of broader applications to astrobiology, these findings suggest that, on the early Earth and planets like Mars, volcanic terrains may have provided transient habitats where life could have established, survived, and potentially left behind detectable biosignatures. Overall, this work demonstrates the rapid microbial colonization of barren environments and the development of stable and complex community structures over time. By analyzing predictable and recognizable patterns underlying microbial succession, this study elucidates how microbial communities can establish themselves in initially harsh environments and underscores the potential for predictive models to be used in ecological research.

## Methods
### Data and Sampling
Initial analysis of lava flow-field emplacement came from Pedersen et al.[25]., Gunnarson et al.[69]., Belart et al.[70]. respectively, which were derived from the European Space Agency Pléiades satellite stereo-images. The digital terrain model (DEM) is from Belart et al.[70]. Boundaries for the old lava flow-fields and respective ages were determined from the Iceland GeoSurvey ISOR GIS server (https://arcgisserver.isor.is/arcgis/rest/services/isor_lokad/sudvestur2016/MapServer). More detailed analysis of lava flow emplacement, including mapping procedures, collection of contextual images, and high resolution orthomosaics can be found on the University of Arizona Research Data Repository[71–73]. Briefly, the lava flow field outlines were mapped at a 1:600 mapping scale. The 2021 outline was based on data collected by the Icelandic Institute of Natural History on September 30, 2021 (0.14 m/pixel). The 2022 flow field outline was mapped using a Loftmyndir orthomosaic captured on August 21, 2022 (0.15 m/pixel). The 2023 flow field outline was mapped using an orthomosaic generated from unmanned aerial system photographic data collected on August 4, 2024 (0.027 m/pixel). Constraints on lava rock ages were determined using a combination of orthomosaics at different dates and field notes. All data products of the sites were annotated and analyzed in ArcGIS Pro (ESRI Inc.; v.3.2.2) with a Lambert Conformal Conical projection.

Starting on 19 March 2021, following a month-long period of intense seismic activity, a 6-month subaerial effusive basaltic eruption began approximately 40 km from Reykjavík, which ended a 781-year period of volcanic dormancy on the Reykjanes Peninsula[25]. Once the Fagradalsfjall eruption began, lava and adjacent soil samples around the lava flow were collected ~biweekly until August of 2021, including lava that had been emplaced the same day of collection. The primary limitation on the time of sampling was the heat of the lava, which would melt through the sterile sample bags unless the lava had had a couple of hours to cool to ambient temperature. Pieces of lava were aseptically broken off the lava flow using an ethanol and flame-sterilized rock hammer and collected into a double-bagged sterile Whirl-Pak. Sterile gloves and facemasks were also used during sampling. The lava temperature in a crack in the rock and the ambient air temperature were recorded using a K-type thermocouple thermometer (Hanna Instruments).

The lava infilled multiple valleys in the Mt. Fagradalsfjall region, which is a complex of glaciovolcanic edifices formed during the Pleistocene Epoch[74]. On 4 April 2021, Vent 2 opened and entered Meradalir Valley (Fig. 1b). On 6 April 2021, lava embayed the base of a mountain with light-toned soil (palagonitized hyaloclastite) particles on its surface, which became the first repeat sampling sites, Sites 1–4 (Fig. 1a). When it became apparent that these sites would be overridden by continued influx of lava into Meradalir, Sites A–C were established at a higher topographic elevation on April 12th, at sites located along inactive margins of lava flow units sourced from Vent 2 (Fig. 1b). Site A was eventually buried near the end of the 2021 eruption in September by a late-state lava breakout during the final days of the eruption. In July 2022, July 2023, and August 2024, samples were collected from Sites B and C again. During the Fagradalsfjall eruption and in subsequent years, post-glacial lava flows (2k, 8k, and 12k year old lavas) were sampled in the immediate vicinity of the Fagradalsfjall flows, as described for the young lava.

On 3 August 2022, a new fissure (Meradalir eruption) opened northeast of the main Fagradalsfjall vent in Meradalir Valley and lasted for 2.5 weeks, overprinting some of the lava from 2021 (Fig. 1a) and ending on 21 August 2022. Samples were collected from this eruption in August 2022 and again from the same site in July 2023 and August 2024.

On 10 July 2023, a third fissure northeast of the previous vents opened near Mt. Litli-Hrútur. The lava flowed south and covered part of Meradalir Valley, including some lava erupted in 2021 and 2022, before ending on 5 August 2023 (Fig. 1a). In July 2023 and again in August 2024, samples were collected from the Litli-Hrútur eruption. Additional samples at different locations around the study area for all three eruptions were also collected in August 2024 to gain insight into the effects of spatial variability on the microbial communities (Supplementary Data 1).

Although all three eruptions were visited by locals and tourists, the risk of contamination was carefully mitigated through strategic site selection and material collection protocols. Public access was only from the southern end of the lava flow field via hiking paths. Therefore, we selected our sampling sites at the northern end of the flow-field to maximize distance from visitor activity. Even following the 2022 eruption, when a trail was rerouted near Sites B and C to provide access to the 2022 vent, our sample collection remained distanced from potential contamination sources. Specifically, the closest trail to any of our sample sites was approximately 150 meters away—a substantial buffer zone. Furthermore, materials were collected several meters inward from the flow margin, an additional measure to ensure samples were obtained from undisturbed areas. While the possibility of human contamination cannot be entirely ruled out, these precautions—coupled with the use of standard protocols—significantly minimized the risk, especially given the highly controlled and methodical approach to sample collection. Careful use of blanks and contamination control during sample processing and data analysis was also used (see DNA Extraction and Sequencing for details).

To determine vectors of microbial colonization of the lava, bioaerosol samples were collected via impaction onto quartz membrane filters (Pall Corporation) using the Deployable Particulate Sampler (SKC, Inc.) with a PM10 selective inlet mounted 1.5 m above the ground. All components of the impactor were cleaned with 70% ethanol and UV-sterilized. Quartz filters were sterilized following an optimized protocol[75]. Briefly, filters were UV-sterilized on both sides for 30 minutes and then individually wrapped in sterilized aluminum foil. The filters were then heated at 500 °C for 8 h and then stored in a UV-sterilized Ziplock bag. Filters inserted into the sampling apparatus, but not exposed to airflow, were used as negative controls. During assembly of the sampling apparatus, sterile gloves and facemasks were used. Samples were collected for 24 h at 15 L/min (for a total of 21,600 L of air per sample). In 2023, a higher volume aerosol sampler was used—the SASS 3100 (Research International, Inc.)—at 300 L/minute for 24 h[76]. After completion of the sampling run, the filtration apparatuses were placed into sterile Whirlpaks. Rain samples were collected using a sterile plastic cup and then filtered onto Sterivex, with a volume of ~50 mL per sample.

Initial sampling of soil was conducted prior to the start of the 2021 Fagradalsfjall eruption, with samples collected within the valleys around Mt. Fagradalsfjall. Additional soil samples were collected throughout the study period in the immediate vicinity of lava sampling sites. Soil samples were collected aseptically by scraping the surface into 15 mL polypropylene tubes using an ethanol and flame-sterilized spoon and then double bagged in sterile Whirlpaks for storage. Hot spring samples were collected from the Krýsuvík Geothermal Area to identify potential sources of thermophiles, with 300 mL of hydrothermal fluid (limited by filter clogging) from high-temperature areas and 1 L from low-temperature areas filtered onto Sterivex. After sample collection, all samples, including lava, were brought back to the University of Iceland on ice and stored at 4 °C (approximately ambient temperature at the time of sampling) until processing the next day or stored at –80 °C. In total 58 lava samples, 3 fumarole samples, 12 old lava samples, 16 soil samples, 10 aerosol samples, 6 rain samples, and 2 hot spring samples were analyzed. All sampling at Fagradalsfjall and sample export were done under permits from the Icelandic Institute of Natural History.

## Environmental Data

Quantitative bulk mineralogy was performed at the X-Ray Diffraction (XRD) Facility, RRID:SCR_022886, in the Department of Chemistry and Biochemistry of the University of Arizona, on a Philips PANalytical X'Pert PRO MPD instrument, using the Cu $K_\alpha$ radiation ($l \approx 1.54$ Å). The analysis of the XRD patterns of the powder (including the assignments and the semiquantitative analysis of the crystalline phases) was performed using the Malvern PANalytical X'Pert HighScore software. The volumetric water content for lava samples was estimated by weighing powdered lava samples, drying at 105 °C for 48 h, and then weighing again. The pH of the lava was measured by making slurries (1:2.5 mass ratio of lava in MilliQ water) and letting it stand for 2 h before measuring the pH of the supernatant. Weather data for multivariate analyses of the microbiome data were obtained from the Festerfjall station (63.85948°N, 22.34358°W) from the Icelandic Meteorological Office. The closest observation to the sampling time and date was added to the sample metadata and concatenated with other physicochemical parameters of the lava samples. Weather data for the aerosols was calculated using an average, maximum, minimum, and upper and lower quartiles for the 24-h sampling period.

## Community-level physiological profiling

Substrate utilization profiling was performed with an EcoPlate™ (Biolog Inc)[13]. One day after sample collection (after storing at 4 °C), slurries of crushed lava, old lava, and soil were prepared by adding 1 g of sample to 20 mL of milliQ (filter sterilized <0.2-μm), vortexing 3× for 1 min and then adding the mixture to the plate. Well-color development was monitored over 7 days at 590 nm and 750 nm (to measure turbidity) using a SpectraMax iD5 plate reader (Molecular Devices). To calculate the average well color development (AWCD), the optical density (OD) readings were first corrected by subtracting OD at 750 nm (to remove turbidity), the OD of the control wells (water), and their initial OD reading at $t_0$[77], with negative values set to zero. Triplicate OD readings for each substrate were then averaged. The substrates were then grouped according to carbon type (carbohydrates, polymers, phenolic compounds, amino acids, and amines)[78] and the AWCD was calculated: $AWCD = \sum \frac{n_i}{s}$ where $n_i$ is the control corrected OD substrate and s is the number of substrate types in that carbon category[79].

## Cell counts

Since the density of basalt is high, microbial cells may not be evenly distributed throughout the rock. Additionally, initial colonization processes likely cause heterogeneous distribution throughout the rock[58]. Consequently, the whole collected lava rocks were crushed into a powder with an autoclaved mortar and pestle and mixed with a sterile spoon to ensure an even distribution of cells[80]. Extraction of cells from the lava powder one day after sample collection (after storing at 4 °C) was

performed following previous work[81]. Briefly, 1 g of lava powder was weighed into a 15-mL polypropylene tube with 10-mL milliQ (filter sterilized <0.2-μm), 1.25-mL of detergent mix (filter sterilized <0.2-μm; 100 mM disodium EDTA dihydrate, 100 mM sodium pyrophosphate decahydrate, 1% vol/vol Tween 80), and 450-μL of 16% PFA. The solution was mixed and allowed to stand for 15 minutes at room temperature, followed by sonication for 3 minutes at 45 KHz, stopping every minute to shake by hand for 30 s and then incubated on ice for 1 h. 1-mL of the solution was then spiked with 100 μL of a solution of 1X SYBR Green I DNA binding dye (Invitrogen, UK), stained for 10 minutes and then filtered onto black 0.2-μm Nuclepore polycarbonate filters (Whatman, UK) using a syringe. The filter was mounted using immersion oil onto a glass slide, covered with a cover slip with additional oil, and then stored at –20 °C until analysis. Microorganisms were enumerated under at least 50 fields of view under 60× magnification on an epifluorescence microscope (Nikon Ti2-E) at the University of Arizona and then converted to cells/g of lava based on the number of fields of view on the filter and the amount of sample input. Blanks were analyzed alongside the samples which consisted of pieces of lava that were UV sterilized on both sides for 30 min, heated at 500 °C for 8 h wrapped in aluminum foil, and then processed as described. No background autofluorescence was observed in the blanks. Samples collected prior to April 2021 were not analyzed for cell counts because of the lack of the necessary reagents in Iceland. Samples that were collected in 2024 were not analyzed for cell counts due to issues with the Sybr Green Dye storage. For old lava, direct cell enumeration using Sybr Green dye was not possible due to autofluorescence and interference from minerals, organic matter, and eukaryotic organisms (e.g., algae fluorescence spilling into the green channel). Consequently, cell abundances for old lava samples were estimated using DNA concentration data under the assumption that bacterial cells contain 8 fg of DNA[82], which has been used in volcanic environments previously[3].

## Culturing and identification of isolated strains

To test for the presence of certain functional groups expected to be involved in successional processes, direct isolation from the lava was attempted following Kelly et al.[16]. Briefly, one day after sample collection (after storing at 4 °C), crushed lava and soil were incubated in a nutrient-rich (tryptic soy broth; TSB) and poor (R2A) broth. To test for the presence of phototrophs, samples were additionally incubated in BG11 broth (pH 7.2). To test for the presence of chemolithotrophs capable of sulfur oxidation, samples were also incubated in sulfur-oxidizer broth (pH 7.2). Finally, to test for the presence of nitrogen-fixing organisms, samples were incubated in nitrogen-free Norris broth (pH 7.2). All enrichment cultures were incubated at 50 °C (to isolate thermophiles) and 11 °C (ambient temperature at the time of collection) for two weeks. The rationale for choosing this combination of media was based on the expectation that phototrophs[83], nitrogen-fixers[10], and chemolithotrophs[16] play key roles in successional processes. TSB and R2A broths were used to capture heterotrophic organisms capable of using a variety of organic substrates. However, the use of predefined cultivation media likely introduced bias towards fast-growing and readily culturable organisms and may have overlooked slow-growing or metabolically novel taxa.

Turbid cultures were streaked onto agar plates with the corresponding nutrient broth and incubated at the appropriate temperature for up to two weeks. Single colonies were re-streaked three times for purification, and isolated strains were grown again in the appropriate nutrient broth and then stored in 25% glycerol nutrient broth at –80 °C until analysis. DNA was extracted using the PureLink Microbiome DNA Extraction Kit according to the manufacturer's instructions and then subjected to 16S ribosomal RNA (rRNA) sequencing using primers 27F (AGAGTTTGATCATGGCT-CAG)/1522R(AAGGAGGTGATCCANCCRCA) in amplification and primer F341 (CCTACGGGAGGCAGCAG) internal of the 16S rRNA gene for Sanger sequencing on an Applied Biosystems 3730XL DNA Analyzer. The obtained sequences were edited and classified, and the closest known relatives of the isolates were identified by BLAST searches in GenBank.

Strains were also subjected to a Schaeffer-Fulton stain to test if the cultures were capable of forming endospores[84].

## DNA Extraction and Sequencing

Contaminant DNA can be present from kit reagents and human commensals on laboratory personnel and in laboratory environments[85]. Nonetheless, cross-contamination is inherent in the sequencing process due to 'tag switching,' or barcode sequencing errors (cross-talk). This is enhanced in low biomass samples, and can make interpretation of the data difficult[86]. To reduce the potential for contamination, enhanced precautions during sample handling were employed, including performing DNA extractions in a specified controlled environment and in a PCR clean hood (AirClean Systems), masking, and working on ethanol-sterilized surfaces. Only one sample was processed per day to reduce the risk of cross-contamination between samples[86]. To increase DNA yield, an optimized extraction protocol was developed, building off previous work[87]. Three extractions with 10 g of crushed lava were performed with the following modifications to the DNeasy PowerMax Soil Extraction Kit (Qiagen): Instead of bead beating on a horizontal vortex adaptor, tubes were homogenized on a FastPrep-24 (MP) bead beater with the BigPrep adaptor for 45 s at 6.5 m/s twice. Tubes were placed on ice between runs and after the bead beating process. The extraction then proceeded according to the manufacturer's instructions until the binding step, where the smaller MB spin columns from DNeasy PowerSoil Pro Kits (Qiagen) were used. The supernatant was kept on ice during the binding process due to long processing times. Two separate spins using 50 μL of elution buffer were used to elute the DNA. The three extractions were pooled and concentrated down to ~50 μL using a vacuum concentrator. Blanks were also processed and consisted of pieces of lava that were UV sterilized on both sides for 30 min, autoclaved, and then heated at 500 °C for 8 h wrapped in aluminum foil.

DNA from the soil and old lava was extracted using the DNeasy PowerSoil Pro Kit (Qiagen) according to the manufacturer's protocol. DNA from the bioaerosol filters was extracted using a modified method[88]. Briefly, cylindrically rolled filters were placed into the preheated (60 °C) lysis buffer from the DNeasy PowerWater Kit (Qiagen). The tubes were then sonicated at 45 KHz at 65 °C for 30 min. After sonication, the tubes homogenized on a bead beater for 45 s at 6.5 m/s twice. To optimize recovery, a syringe was placed inside of a 50 mL polypropylene tube and then lysate was added to the inside of the syringe and centrifuged at 1000 g for 4 minutes to collect liquid from the filter debris[75]. Afterward, the extraction proceeded according to the manufacturer's protocol. The elution volume was 75 μL. Field blanks were also processed the same way. DNA from the hydrothermal fluids and rain was extracted using the DNeasy PowerWater Sterivex Kit (Qiagen) following manufacturer protocols. Blanks consisted of unused Sterivex filters. DNA from all extraction methods was quantified using a Qubit Fluorometric Quantification System (Thermo Fisher Scientific) with the high-sensitivity assay for double-stranded DNA.

The 16S rRNA gene v4 variable region primers 515 F (5'-GTGYCAGCMGCCGCGGTAA-3') and 806 R (5'-GGACTACNVGGGTWTCTAAT-3') were used in a 35 cycle PCR using HotStarTaq Plus Master Mix Kit (Qiagen) under the following conditions: 95 °C for 5 min, followed by 35 cycles of 95°C for 30 s, 53 °C for 40 s and 72 °C for 1 min, after which a final elongation step at 72 °C for 10 min was performed. PCR products were then checked in a 2% agarose gel to confirm amplification. Samples were then multiplexed using unique dual indices and pooled in equal proportions based on molecular weight and DNA concentration. The pooled samples were purified using calibrated Ampure XP beads. Sequencing was performed at MR DNA (Shallowater, TX, USA) on a MiSeq using paired-end 2×250 bp chemistry following the manufacturer's guidelines.

Sequence data was processed in QIIME2's Python API (v2024.10)[89]. Sequences were demultiplexed and primers were removed using Cutadapt[90] and then denoised, chimeric sequences were removed, and truncated using DADA2[91]. For the different sequencing runs, we performed these steps separately and combined the feature tables for downstream analysis. We also compared beta diversity of the samples to ensure no clustering occurred

according to the sequencing run. A reference database was built using RESCRIPt[92] on the SILVA database (v138.1)[93,94]. Representative sequences for each amplicon sequence variant (ASV) were used for taxonomic assignment using the built database[95]. A phylogenetic tree was created using MAFFT alignment[96] and FastTree[97].

The dataset was manually screened for known contaminants[87,98,99], including comparing relative abundances against respective blanks, and sequences identified as *Staphylococcus*, *Finegoldia*, and *Streptococcus* were removed. While it is possible these sequences were not contaminants, we chose to be more stringent given the low-biomass nature of our samples. Additional decontamination was conducted using the package decontam integrated in QIIME2[100] for each independent sequencing run, which has been shown to be effective in identifying contaminants when the potential sources are poorly defined[101]. In the prevalence method, decontam uses the prevalence of each sequence feature in true samples compared to corresponding negative controls to the specific DNA extraction technique (e.g., PowerSoil, Sterivex, etc.). Any ASV with a final $p$-value < 0.1 was suspected to be a contaminant and removed (Supplementary Figs. 10, 11). Mitochondrial and chloroplast sequences were removed. After all the processing steps, the minimum number of sequences was 10,720, which was chosen as the rarefaction depth for alpha and beta diversity analysis, and excluded sample L206 in some analyses. In total, 60,952,834 raw reads were generated across all samples and sequencing runs. Rarefaction curves showed that the sequencing depth was adequate to capture the taxonomic diversity in most of the samples, except for two soil samples. Subsequent alpha and beta diversity analyses were completed using the rarefied tables.

## Statistics and reproducibility

General statistics (e.g., linear regressions and $R^2$) were conducted in Python (v3.10) using the pandas (v2.2.2) and NumPy (v1.26.4) libraries unless otherwise stated. Alpha diversity using Faith's phylogenetic diversity (PD)[102,103] and beta diversity using weighted Unifrac[104,105] and Jaccard distances and associated principal coordinate analysis were analyzed using QIIME2's core-metrics-phylogenetic command. To calculate the weighted Unifrac and Jaccard distances between sequential states, QIIME2's q2-longitudinal plugin with the command first-distances was used[106] using the order of sample collection as the state variable as a proxy for lava rock age and site name as the individual ID.

To estimate the proportions of various sources of microorganisms colonizing the lava flow, we used the Bayesian algorithm SourceTracker2 (v2.0.1) on the rarefied data, which uses a Gibbs sampler, a type of Markov chain Monte Carlo algorithm[23]. Source categories were soil, aerosol, old lava, hot springs, rain, and unknown. Briefly, the algorithm works in four steps: (1) Randomly assign the sequences in the lava to source environments. (2) Select a sequence from Step 1, calculate the probability that this sequence originated from any of the source environments, and then update the assigned source environment of the sequence based on the calculated probabilities. (3) Repeat Step 2 many times, and at set intervals during the iterations, record the source environment assignments of all the sequences in the lava. (4) Calculate the overall probability of the assignments from Step 3, and the standard deviation, and then move to the next lava sample.

Processes dictating community assembly were determined using phylogenetic null modeling[45,47,48,107]. Phylogenetic turnovers (shifts in composition weighted by the phylogenetic similarity between taxa), and turnover expected by chance (using null models) were used to estimate if the development of the microbial community in the lava flow was influenced by deterministic or stochastic processes. Deterministic factors include environmental conditions that select species capable of survival, resulting phylogenetic clustering. Deterministic processes also include competitive exclusion. Stochastic processes, in contrast, are a result of dispersal limitation, homogenous dispersal, and drift. To disentangle these mechanisms, the trans_nullmodel class in the microeco R package[108] was used. Mantel correlograms were used to determine if there were significant phylogenetic signals at short phylogenetic distances (e.g., if similarity among ASVs is higher or lower than expected by chance). Significant positive correlations

with environmental data indicate that ecological similarity among ASVs is higher than expected by chance. While there are likely slight environmental differences between sites in the eruption study area, weather conditions at the site level were not resolved in this study, and parameters such as pH and water content remained relatively constant (Supplementary Data 1). Consequently, due to the homogeneity of these measured physicochemical parameters through time and space in various sites, we took age to be a proxy for cumulative environmental effects in the context of determining phylogenetic signal. If there are significant trends, the data can be analyzed for phylogenetic turnover patterns, which include the net relatedness index (NRI), the nearest taxon index (NTI), beta nearest taxon index (βNTI), beta net relatedness index (βNRI), and Bray-Curtis-based Raup-Crick (RCbray).

To characterize if the alpha diversity of the lava samples was governed by environmental filtering or overdispersion, we used NRI and NTI. NRI is calculated as the mean pairwise phylogenetic distance of taxa in a single sample and quantifies the overall clustering of taxa in a tree. It is calculated as

$$NRI = -(mn(X_{obs}) - mnX(n)/sdX(n)) \qquad (1)$$

where $X_{obs}$ is the phylogenetic distance between two taxa, $mn(X_{obs})$ is the pairwise mean distance of all (n) taxa, and $mnX(n)$ and $sdX(n)$ are the mean and standard deviation respectively based on a null model consisting of shuffling the tree branches in the sample pool over 10,000 iterations. Additional information about the clustering and associations of taxa within a sample can also be gained by calculating NTI, which is a measure of phylogenetic distance to the nearest taxon on the tree. It is calculated as

$$NTI = -(mn(Y_{obs}) - mnY(n)/sdY(n)) \qquad (2)$$

where $Y_{obs}$ is the phylogenetic distance to the nearest taxon and the other parameters are calculated as in Eq. 1[109].

Next, the abundance weighted phylogenetic distance between pairwise samples (beta mean pairwise distance; βMPD) was computed with 10,000 iterations of null models. βNRI was then computed by calculating the difference between the observed βMPD and the mean of the βMPD null models divided by the standard deviation of the null models. βNRI therefore represents the standardized effect size of phylogenetic turnover between samples, providing a measure of whether community composition between samples is phylogenetically clustered more or less than expected under random assembly processes. Similarly, beta mean nearest taxon distance (βMNTD) was computed which is the pairwise abundance weighted mean phylogenetic distance among closest relatives. βNTI was then computed by calculating the difference between the observed βMNTD and the mean of the null distribution[45,48].

Using the command cal_process in microeco, the fraction of pairwise comparisons was divided into categories. These were divided into age bins to observe the inferred processes affecting community assembly as a function of lava age. βNRI or βNTI < –2 or >+2 indicates that the phylogenetic distance deviates from the null by more than two standard deviations. Scores less than −2 indicate that the observed phylogenetic turnover was lower than the null model, suggesting that homogeneous selection between the compared communities causes higher than expected phylogenetic similarity. In other words, constant environmental factors are inferred to have caused low turnover in the community. Conversely, scores greater than +2 indicate the dominance of variable selection, meaning shifts in environmental factors caused high community turnover. Scores between –2 and +2 indicate that the community was dominated by stochastic processes. In this regime, RCbray gives further insight into whether the compositional turnover was governed by dispersal limitation (limited exchange of organisms between communities allowed community composition to diverge due to drift), homogenous dispersal (dispersal was high enough to cause low turnover by overwhelming deterministic processes), and ecological drift acting alone (fluctuations in population due to chance events)[45,47,110]. Similar to βNRI and βNTI, RCbray is based on the comparison between observed and expected levels of turnover based on a null model, but without

phylogenetic information, only species relative abundance. RCBray calculates the magnitude of deviation between Bray-Curtis and Bray-Curtis expected under randomization using 10,000 null models. RCBray >0.95 represents dispersal limitation combined with drift, RCBray <−0.95 represents homogenous dispersal, and |RCBray| < 0.95 represents drift acting alone[107]. To verify that the inferred assembly processes were not unduly influenced by unresolvable site-specific environmental differences, we performed these analyses on subsets of the data, excluding individual sites in a manner analogous to leave-one-out cross-validation (Supplementary Fig. 7B).

There are several limitations to these methods since inferring ecological processes based on phylogenetic metrics alone is challenging. The phylogenetic signal may not be true in all cases on the tree of life[111]. Moreover, some have argued that competition could lead to phylogenetic clustering[112]. Finally, this null modeling approach forces stochastic and deterministic processes to trade off. Discrete processes may not always be the case, since a single mechanism can yield both deterministic and stochastic patterns[61].

Linear mixed effects models to account for the spatial variability of the sites were applied to Faith's PD, relative abundance of certain taxa, NRI, and NTI. Linear mixed effects models applied to longitudinal studies permit assessing changes in response variables over time while accounting for fixed and random effects as well as variations in the timing and number of observations. Fixed effects model systematic patterns (e.g., treatment conditions) while random effects vary over the individuals or sites under study[34]. In this study, we treat the sampling site as the random effect because it was not part of the original study design and not systematically chosen. The LME model was conducted using the Python package pymer4[113] to generate a slope and y-intercept. After fitting the model, the between-site (random intercept) variance component $\sigma_0^2$ and the residual (within-site) variance component $\sigma^2$ were calculated. The Intraclass Correlation Coefficient (ICC) was calculated as the ratio of the between-group variance to the total variance:

$$ICC = \frac{\sigma_0^2}{\sigma_0^2 + \sigma^2} \qquad (3)$$

ICC provides a measure of how strongly units within the same site resemble each other, with values closer to 1 indicating a higher proportion of variance attributable to differences between sites. LME models were applied to samples <100 days old (when biweekly sampling was occurring) and then again over the course of the entire study period. Important features were identified using QIIME2's longitudinal plugin[106] with the feature-volatility command, plotted versus time, and then subjected to the LME analysis. Faith's PD over the course of thousands of years (e.g., including the old lava) was analyzed with Spearman's rank test using QIIME2's alpha-correlation because there are repeat measures that violate the assumption of an LME model.

Lava rock ages were predicted based on microbial community data (e.g., ASV abundances) using an ensemble learning method. This method was implemented using QIIME2's q2-sample-classifier plugin with commands regress_samples and then predict_regression[114]. We split the data into training (2021 Fagradalsfjall eruption) and test (2022 Meradalir and 2023 Litli-Hrútur eruptions) sets. The rationale for splitting data by eruption event was to ensure that the model learned patterns from one eruptive event and then tested its predictive ability on data from subsequent events. Of the various machine learning regressor methods we explored, including lasso, ElasticNet, linear support vector regression, and ridge regression[115], the random forest regressor quickly yielded the best results in terms of more accurately predicting lava rock age and thus was used to train the final model. Because our target variable (timepoint) is independent of site-specific effects and not a fixed site-level outcome, using a random forest regressor to predict lava rock age avoids the information leakage concerns associated with longitudinal data[116] and is consistent with established approaches for predicting temporal variables[65].

Hyperparameters were optimized (parameter_tuning=True) to improve model performance. The final model used a sufficiently large number of trees (e.g., 1000) to ensure robust estimates of feature importance and to reduce variance. Moreover, to minimize problems due to overfitting, the model optimized input feature selection using recursive feature elimination to determine the most important predictive features (optimize_feature_selection=True). The training procedure included k-fold cross-validation, providing a reliable internal estimate of model generalization performance. Performance was primarily evaluated using the coefficient of determination ($R^2$) to quantify the proportion of variation in the true ages explained by the predicted ages. Mean Absolute Error (MAE) and Root Mean Squared Error (RMSE) were also calculated to assess the average magnitude of prediction errors. A paired $t$-test comparing the predicted and actual ages determined whether systematic biases (e.g., consistent overestimation) were statistically significant. Models were trained and evaluated using features aggregated at various taxonomic levels, but order-level features yielded the best predictive performance.

To determine if pathways and metabolisms could be predictive of successional processes, we used the paprica (v0.8.0)[50], PICRUSt2 (v2.6.2)[51], and FAPROTAX (v1.2.11)[52] metabolic inference pipelines. Feature tables were exported from QIIME2 and run using default parameters for each model. The output of paprica and PICRUSt2 are estimates of abundances metabolic pathways, whereas FAPROTAX maps ASVs to broad ecological functions based on implicit assignments of a trait to taxa. The paprica pipeline depends on EPA-ng[117], Gappa[118], Infernal[119], and Pathway Tools[120] and PICRUSt2 depends on EPA-ng and Gappa. Metabolic pathway results were then converted to biom format[121] and then re-imported to QIIME2 as feature tables for analysis with q2-sample-classifier with metabolic pathways as predictive features rather than ASVs or collapsed taxonomic compositions. Results must be interpreted with caution since FAPROTAX assumes functional uniformity within taxa, while PICRUSt2 and paprica rely on reference genomes, which can result in errors, especially for extreme environments with low representation in genome databases and the literature. PICRUSt2 was additionally used to infer 16S rRNA copy number and LME models were applied to observe the variation with lava rock age.

### Reporting summary

Further information on research design is available in the Nature Portfolio Reporting Summary linked to this article.

### Data availability

Raw sequences are available in the GenBank Sequence Read Archive under accession numbers SAMN46055121–SAMN46055246, BioProject accession number PRJNA1205709. GenBank accession numbers for the strains isolated during this work are PQ835131–PQ835142. Environmental covariates are available in Supplementary Table 1 and Supplementary Data 1. Data for plots presented in the main manuscript are available in the Supplementary Data 2. Samples and any remaining information can be obtained from the corresponding author upon reasonable request.

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

## Acknowledgements

Funding support was provided by the National Science Foundation RAPID awards 2128606 to S.D. and 2131889 to C.W.H., National Defense Science and Engineering Graduate (NDSEG) Fellowship Program to N.H., Geological Society of America Graduate Student Grant 13183-21 to N.H., Lewis and Clark Fund for Exploration and Field Research in Astrobiology from the American Philosophical Society to N.H., The University of Arizona Graduate and Professional Student Council Travel Grant to N.H., Arizona Astrobiology Center Seed Grant to S.D. and N.H. and Scialog (Heising–Simons Foundation) grant 2023-4652 to C.W.H. We thank Dr. Andrei Astashkin (University of Arizona – CBC XRD Facility, RRID:SCR_022886) for the XRD data collection and analysis. We thank Madison Tuohy, Matthew Varnam, Joana Voigt, Peter Schroedl, Brett Carr and colleagues at the University of Iceland for assistance in the field. We thank Andri Stefánsson at the University of Iceland for use of his plate reader. We also thank Abigail Boatwright, Brooke Rabe and Dean Billheimer for helpful conversations regarding the statistical analyses.

## Author contributions

S.D. designed research; N.H., S.D. and C.W.H. performed research; N.H., C.W.H., S.B. and S.D. contributed new reagents/analytic tools; N.H. analyzed data; and N.H. wrote the original draft and C.W.H., S.B., and S.D. reviewed and edited the manuscript.

## Competing interests

The authors declare no competing interests.
