## [Transparent Peer Review file · Communications Biology]

Three Eruptions at the Fagradalsfjall Volcano in Iceland Show Rapid and Predictable Microbial Community Establishment

Corresponding Author: Dr Solange Duhamel

Version 0:

Reviewer comments:

Reviewer #1

(Remarks to the Author)

General comments:

This manuscript focuses on investigating the colonization processes and changes in microbial community by a stochastic natural event. This research is very original, and the experimental design is good. The statistical methods employed are robust and sufficient to address the research questions. I, therefore, recommend this manuscript to be published, after the authors address the following questions and concerns.

Specific Comments (Please see the line number annotated document):

Introduction:

The background information connects well with the study. Research questions are well-defined and relevant.

L27: "uninhabited to inhabited". The words (un)inhabited and (un) habitable have distinct meanings. For example, inhabited means a place that currently has people or other living organisms residing in it. Habitable means a place is suitable for living, but may or may not have inhabitants. I suggest the authors go through the manuscript and ensure these terms are used correctly.

L35: "C9)". Consider moving the citation 9 after the parenthesis.

L44-48: Major volcanic eruptions and the formation of volcanic islands are rare events, even in volcanically active regions like Iceland. The authors should provide more background information regarding the volcanic island of Surtsey in Iceland, formed from 1963 to 1967.

L47-48: It would be helpful to include some details about the diverse microbial community, such as the types of microorganisms present.

Results:

In general, detailed information and descriptions related to the methods should be moved to the Methods section.

L99: "...so DNA concentration was therefore converted to cell counts". This is related to the methods and should be moved to the "Methods" section.

L132: "(Supplementary Fig. 3)". Supplementary Fig. 3 shows the change in microbial community composition with lava rock age. This is one of the most important findings of the study and should not be placed in the supplementary materials. The authors should move this figure into the main body of the manuscript.

L255-259: These sentences explain what the NTI and NRI indices measure and why these indices were used. I recommend moving these to the Methods. The authors should focus on reporting the results of the NTI and NRI calculations

Discussion

Overall, the discussion is sufficiently thorough.

L367: "Fimmvörðuháls in Iceland³⁶ and Eremiobacterota in Hawai'i". Please check the spelling of "Fimm.." and "Hawaii".

Methods:

Generally, more information is needed to clarify some methods.

L475: "Lava Sampling...". The authors provided Figure 1 (a map) showing the sampling locations. I suggest including a supplementary table that lists the GPS coordinates for each sampling site.

L697-699: The sequencing method used in this study is unclear. Was MiSeq 2×250 bp or 2×300 bp sequencing performed?

L701: "(v2024.10z)": Please check the QIIME2 version. The QIIME 2 team typically uses numerical versioning, such as '2024.10'.

L721-723: "After all the processing steps, the minimum number of sequences was 10,720 which was chosen as the rarefaction depth for alpha and beta diversity analysis..." Please include the total number of raw reads obtained after sequencing.

L739: "SourceTracker2". Is SourceTracker2 a standalone software, or is it part of the QIIME 2 software package? Please include the version number and appropriate citation for SourceTracker2.

Reviewer #2

(Remarks to the Author)

The manuscript by Hadland et al describes a unique natural experiment where the establishment and succession of microbial communities associated with newly exposed lava (which was presumed to be initially sterile) were monitored. Although the study was entirely descriptive, I think the thoroughness and thoughtfulness of the natural experiment make the results very relevant and interesting for most microbial ecologists. The fortuitous replication of the observations allowed the authors to employ what I consider to be probably the most appropriate use ever of random forest regression in a microbial ecology study (even though it only worked until the main source of the microorganisms switched, which perhaps made it unexpectedly damning evidence that this approach has limited utility ultimately). The results are predictable but nevertheless very interesting from the perspective of priority effects (or the apparent lack thereof). I think the manuscript is absolutely acceptable for publication after some very minor improvements, and I encourage the authors to consider a couple of points that they hadn't explicitly mentioned.

Specific comments:

- Before microorganisms associated with rain washed everything else out at the one-year mark, the apparent lack of priority effects (i.e., "predicable [sic] nature of microbial colonization"), which the authors attributed to the harsh environments, contradicts the complex and spatially heterogeneous microbial biogeography observed in many arid (i.e., harsh) habitats. Perhaps this is a function of time, and priority effects manifest over much longer timeframes than covered in this study, or perhaps newly cooled lava imposed such strong selective forces that there is only one straight and narrow path for primary succession. Another question is why the 'rain microbiome' apparently arrested the succession processes, at least within the timeframe of the study. At what point does new lava begin to resemble old lava microbiologically, and what factors may trigger that shift? I would like to see the authors' thoughts on these points.
- Why did the authors think that a regression based on PAPRICA outputs would increase the generalizability of the model? 'Predicted' metagenomes from PAPRICA, PICRUST, or Tax4Fun by nature are based on correlations and cannot introduce new independent variables.
- Please refer to relevant figures and tables in the Results section consistently (e.g., line 187).
- The authors should do a spellcheck and a careful proofread. See [sic] above.
- Please italicize all Latin taxonomic names as per <https://journals.asm.org/writing-your-paper#nomenclature>
- Some of the symbols in Figure 1a are too small, and it is unclear how many there are for each type. Figure 1b is quite difficult to read--consider making the labels bright yellow.

Reviewer #3

(Remarks to the Author)

This manuscript presents a comprehensive and well-executed examination of microbial colonization on newly emplaced lava flows in Iceland. The authors leverage the 2021-2023 Fagradalsfjall eruptions as a natural experiment to document the transition from sterile substrate to inhabited ecosystem with unprecedented temporal resolution - from hours to years post-emplacment. The multi-faceted approach combining 16S rRNA sequencing, cell counts, culturing, and statistical modeling provides robust evidence for a predictable two-stage colonization process.

The study's major strengths lie in its exceptional temporal resolution, the fortuitous natural replication provided by three sequential eruptions, and the integration of multiple data streams to build a comprehensive picture of community assembly. The random forest model's ability to predict successional stage across eruptions with high accuracy ($R^2 = 0.95$) is particularly compelling and demonstrates the predictable nature of these assembly processes.

While the authors have thoughtfully addressed spatial variability by treating site as a random effect in their linear mixed effects models, I wonder if there might be systematic differences in environmental parameters between sampling locations that could influence colonization patterns beyond what's captured. The authors note that pH and water content remained relatively constant, but could there be gradients in lava chemistry across flows or systematic differences in exposure to surrounding microbiomes? The spatial distance from old to new lava spans potentially distinct soil/water/air microbiomes that could create a bias in what's emigrating to the new surfaces. Similarly, while all lava flows presumably start chemically similar, subtle differences between eruption events could create environmental biases that compound with temporal effects.

The disparity between community structure and metabolic function predictability ($R^2 = 0.95$ vs 0.64) is expected but intriguing. As the authors acknowledge, this likely stems from both sparse metabolic annotation data and the physiological plasticity/functional redundancy inherent in microbial communities. This gap between "who's there" and "what they're doing" remains one of the field's persistent challenges.

What I find most compelling is the observed decrease in NRI over time, suggesting a fundamental shift in assembly drivers.

The initial harsh conditions of fresh lava create strong abiotic filters - only the hardiest arrive and survive. But as time progresses and community coalescence increases, biotic interactions increasingly shape assembly. The emigrating microbes from surrounding environments must now contend with an established lava surface community, making competition, resource partitioning, and other biotic factors increasingly important. This reminds me of priority effects in classical ecology - who gets there first matters.

This pattern contrasts interestingly with other primary succession studies. Nemergut et al.'s (2016) beach microcosm work found decreases in rRNA operon copy number during succession, inferring a shift from r-selected early colonizers to K-selected later arrivals. But their system was nutrient-rich - quite the opposite of these harsh lava flows. I wonder if the authors' data might reveal an inverted pattern? Perhaps the extreme conditions initially favor K-selected organisms capable of surviving stress and efficiently using limited resources, rather than fast-growing opportunists. Examining rRNA copy number patterns could provide fascinating insights into colonization strategies.

The increasing influence of community coalescence with time deserves deeper exploration. As succession progresses, the introduction of new species from neighboring areas coupled with an established community should create a complex mosaic of interactions. Rather than a simple, predictable successional trajectory, we might expect increasing spatial and temporal heterogeneity as these mixing processes intensify. Yet remarkably, the authors still find predictable patterns - perhaps testament to the overriding influence of the harsh lava environment even as coalescence increases.

A few additional considerations: The ephemeral fumarole communities remaining arrested in an early successional state despite being one year old is particularly intriguing. This suggests that even modest environmental stress can override temporal progression, keeping communities in a perpetual pioneer state. Also, the shift from aerosol/soil sources to rain-derived communities after the first winter represents a fundamental change in colonization dynamics that warrants further investigation.

In summary, this work makes significant contributions to our understanding of microbial primary succession in extreme environments. The documentation of the transition from abiotic to biotic assembly drivers, replicated across multiple eruptions, provides rare empirical support for theoretical predictions about community assembly. The predictive modeling demonstrates that microbial communities can serve as reliable indicators of ecosystem development stage. With some expanded discussion of environmental covariates, functional predictions, and the complex interplay between harsh conditions and community coalescence, this manuscript will be of broad interest to microbial ecologists and those studying ecosystem development under extreme conditions.

Version 1:

Reviewer comments:

Reviewer #1

(Remarks to the Author)

The manuscript has been significantly improved after revision. The writing is now clear, and the research is novel. The authors have addressed all the questions from the first review. Therefore, I recommend that the editor accept this manuscript.

Reviewer #2

(Remarks to the Author)

The authors have addressed my comments adequately.

Reviewer #3

(Remarks to the Author)

The authors adequately addressed my three main concerns: (1) expanded discussion of environmental heterogeneity and site variability effects, (2) added multiple functional prediction tools with appropriate limitations caveats, and (3) enhanced analysis of priority effects and succession dynamics including the requested rRNA copy number data.

****Author responses are in red**

* The reviewers ask for more details especially in the methods (and partly a more detailed results/discussion), we want to note that we strive for the highest possible reproducibility and allow unlimited space for methods

We have added more detail to the methods, detailed below in response to the reviewers.

**** R2 has two very important questions on the succession of microbial communities (also raised/commented on by R3) and more importantly (also R3 commented on the function predictability), on the use of the PAPERICA tool. We want to point out that the use of metabolic inference is only encouraged when compared with other tools and after mentioning the limitations (which is only partly already done in your manuscript), for this see**

also <https://www.microbiologyresearch.org/content/journal/mgen/10.1099/mgen.0.001203><<https://url.usb.m.mimecastprotect.com/s/jBjYCDwKGDTOQEDApCWfPijFUf?domain=microbiologyresearch.org>>

and <https://microbiomejournal.biomedcentral.com/articles/10.1186/s40168-020-00815-y><<https://url.usb.m.mimecastprotect.com/s/ut3UCEKLJEt6KvkxyTwh2i751G6?domain=microbiomejournal.biomedcentral.com>>

We agree that functional prediction tools are inherently limited by their dependence on curated databases and may not introduce truly independent variables. We do not explicitly discuss the outputs (e.g., specific metabolic pathways) of these tools. Rather, the goal was to demonstrate that the predicted pathways, despite their inferential nature, may still reflect ecological patterns and offer complementary insight into the potential functional shifts associated with microbial succession.

To address this, we revised the manuscript in two ways:

- 1. We expanded our random forest regression analysis to compare predictions based on multiple functional inference tools (adding PICRUSt2 and FAPROTAX) alongside the original models. This approach allowed us to evaluate consistency and robustness across tools and to explore how different assumptions shape predictive performance.**
- 2. We explicitly discussed the limitations of using such tools in a predictive framework, referencing the recommended literature.**

While our results show that functional predictions can contribute to explaining some variation in succession patterns, we acknowledge that these results are correlative and

limited by the quality and resolution of available reference genomes and database curation. The new paragraph in the results now reads (Page 14–15, Lines 301–312):

“To explore whether inferred metabolic trait shifts could predict lava rock age, predicted pathway abundances for each sample using the paprica⁴⁸, PICRUST2⁴⁹, and FAPROTAX⁵⁰ tools were used (Fig. 6B), as done previously for predicting biogeochemical processes⁵¹ to provide a complementary view of community potential. These models had a lower performance than the absolute abundance data (paprica: $R^2 = 0.64$; PICRUST2: $R^2 = 0.69$; FAPROTAX: $R^2 = 0.79$) and had an overestimation bias (paprica: MAE = 188.8, RMSE = 225.2; PICRUST2: MAE = 136.0, RMSE = 171.9; FAPROTAX: MAE = 314.8, RMSE = 388.1). While metabolic inference tools must be treated with caution due to their reliance on curated databases, cultured isolate characterization, or reference genomes^{52,53}, the inclusion of inferred metabolisms nonetheless serve as a valuable exploratory approach and foundation for hypothesis generation regarding functional shifts over time.”

Additionally, Figure 6B was updated (below) to include the multiple regressions and the methods (Page 43 Lines 935–950) were updated to reflect the addition of the other tools.

** Related to that, while not covered specifically by any of the referees, we would like to

point out that the use of cultivation medium for the isolates is crucial and should be discussed better: Why was this medium used and what are the limitations?

Multiple types of cultivation media were used at different temperatures to capture different types of functional groups expected to be involved in successional processes. We have tweaked the methods statements to include references that justify their use and included a statement on limitations (Page 31, Lines 679–685):

“The rationale for choosing this combination of media was based on the expectation that phototrophs⁷⁹, nitrogen-fixers¹⁰, and chemolithotrophs⁷⁸ play key roles in successional processes. TSB and R2A broths were used to capture heterotrophic organisms capable of using a variety of organic substrates. However, the use of predefined cultivation media likely introduced bias towards fast-growing and readily culturable organisms and may have overlooked slow-growing or metabolically novel taxa.”

** As R3 pointed out, your study does not take into account potential confounders/environmental variables of e.g. lava chemistry beyond pH/water content. Thoroughly discuss and mention limitations.

We discuss these points thoroughly in response to R2 and R3's comments. Briefly, environmental differences were largely captured by treating site as a random effect in our linear mixed effects models. Major deviations would also have been apparent in the random forest models, but we still find consistent shifts in microbial community assembly regardless of eruption, likely because of environmental filtering and potentially because the timescale of the study was too short to capture the compounding effects of micro-environmental heterogeneity. We are currently working on follow up studies to quantitatively link environmental heterogeneity to microbial biomass and diversity to capture these nuances.

For all graphs depicting a single point value (e.g., mean) with error bars, you must add individual data points or convert the graph to a boxplot or dot-plot to show data distribution.

Where appropriate, we have already done so in our plots.

It's mandatory to provide access to the numerical source data for graphs and charts either through a repository or by providing the data in a Supplementary Data file (in excel format).

We have included an Excel file with the source data for the plots in the main text.

Please ensure that you have complied with the data deposition policies at the Nature Portfolio, please see here<<https://url.usb.m.mimecastprotect.com/s/yW0rCGwNLJTL0D24nTpiEiBnKk6?domain=nature.com>>.

Our data has been uploaded to GenBank or the University of Arizona Research Data Repository (for mapping data) and downstream data products are included as a supplementary Excel file.

Reviewers' comments:

Reviewer #1 (Remarks to the Author):

General comments:

This manuscript focuses on investigating the colonization processes and changes in microbial community by a stochastic natural event. This research is very original, and the experimental design is good. The statistical methods employed are robust and sufficient to address the research questions. I, therefore, recommend this manuscript to be published, after the authors address the following questions and concerns.

Specific Comments (Please see the line number annotated document):

Introduction:

The background information connects well with the study. Research questions are well-defined and relevant.

L27: “uninhabited to inhabited”. The words (un)inhabited and (un) habitable have distinct meanings. For example, inhabited means a place that currently has people or other living organisms residing in it. Habitable means a place is suitable for living, but may or may not have inhabitants. I suggest the authors go through the manuscript and ensure these terms are used correctly.

We double checked this terminology throughout the manuscript and did not find any issues with respect to the definitions provided here.

L35: “C9)”. Consider moving the citation 9 after the parenthesis.

Done.

L44-48: Major volcanic eruptions and the formation of volcanic islands are rare events, even in volcanically active regions like Iceland. The authors should provide more background information regarding the volcanic island of Surtsey in Iceland, formed from 1963 to 1967.

L47-48: It would be helpful to include some details about the diverse microbial community, such as the types of microorganisms present.

We have added additional details about Surtsey and subsequent microbial investigations. The section now reads (Pages 2–3, Lines 44–52):

“Even the volcanic island of Surtsey in Iceland, formed from 1963 to 1967 during a major submarine eruption and designated as a nature reserve to observe succession, was not comprehensively surveyed for microorganisms until decades after the eruption. Preliminary surveys were conducted shortly after the island’s formation and did potentially detect early colonizers such as *Cyanobacteriota* and *Actinomycetota*¹⁹. By the time broader microbiological investigations were conducted, Surtsey already hosted a diverse microbial community including heterotrophs and members of *Enterobacteriaceae* associated with vegetated areas and bird colonies^{20,21}.”

Results:

In general, detailed information and descriptions related to the methods should be moved to the Methods section.

L99: “...so DNA concentration was therefore converted to cell counts”. This is related to the methods and should be moved to the “Methods” section.

This sentence was deleted and the section now reads (Page 5, Lines 103–105):

“Old lavas were estimated to be around $\sim 10^9$ cells/gram regardless of the $\sim 10,000$ -year age difference between old lava flow sites (Supplementary Fig. 1B).”

L132: “(Supplementary Fig. 3)”. Supplementary Fig. 3 shows the change in microbial community composition with lava rock age. This is one of the most important findings of the study and should not be placed in the supplementary materials. The authors should move this figure into the main body of the manuscript.

We have moved this figure into the main text and updated references to figure numbers throughout the text.

L255-259: These sentences explain what the NTI and NRI indices measure and why these indices were used. I recommend moving these to the Methods. The authors should focus on reporting the results of the NTI and NRI calculations

This sentence was made more concise and now reads (Page 12, Lines 259–261):

“Under the assumption of a phylogenetic signal, ecological processes governing the phylogenetic structure of individual lava samples were investigated using the nearest taxon index (NTI) and the net relatedness index (NRI).”

Discussion

Overall, the discussion is sufficiently thorough.

L367: “Fimmvörðuháls in Iceland³⁶ and Eremiobacterota in Hawai‘i”. Please check the spelling of “Fimm.. “ and “Hawai‘i”.

We have double-checked the spelling of Fimmvörðuháls and confirmed that it is correct. We use the okina in “Hawai‘i” to reflect the correct Hawaiian spelling.

Methods:

Generally, more information is needed to clarify some methods.

L475: “Lava Sampling...”. The authors provided Figure 1 (a map) showing the sampling locations. I suggest including a supplementary table that lists the GPS coordinates for each sampling site.

GPS coordinates for the sites can be found in Supplementary Table 2 and were uploaded as metadata in GenBank.

L697-699: The sequencing method used in this study is unclear. Was MiSeq 2×250 bp or 2×300 bp sequencing performed?

2x250 bp chemistry was performed. The methods statement now says (Page 34, Lines 746–748):

“Sequencing was performed at MR DNA (Shallowater, TX, USA) on a MiSeq using paired-end 2x250 bp chemistry following the manufacturer’s guidelines.”

L701: “(v2024.10z)”: Please check the QIIME2 version. The QIIME 2 team typically uses numerical versioning, such as '2024.10'.

This was a typo. The version used was v2024.10.

L721-723: “After all the processing steps, the minimum number of sequences was 10,720 which was chosen as the rarefaction depth for alpha and beta diversity analysis...” Please include the total number of raw reads obtained after sequencing.

The section now reads (Page 35, Lines 770–774):

“After all the processing steps, the minimum number of sequences was 10,720 which was chosen as the rarefaction depth for alpha and beta diversity analysis, and excluded sample L206 in some analyses. In total, 60,952,834 raw reads were generated across all samples and sequencing runs.”

L739: “SourceTracker2”. Is SourceTracker2 a standalone software, or is it part of the QIIME 2 software package? Please include the version number and appropriate citation for SourceTracker2.

SourceTracker2 is a standalone software (<https://github.com/caporaso-lab/sourcetracker2>) and the citation was already included. We added the version number (v2.0.1).

Reviewer #2 (Remarks to the Author):

The manuscript by Hadland et al describes a unique natural experiment where the establishment and succession of microbial communities associated with newly exposed lava (which was presumed to be initially sterile) were monitored. Although the study was entirely descriptive, I think the thoroughness and thoughtfulness of the natural experiment make the results very relevant and interesting for most microbial ecologists. The fortitous replication of the observations allowed the authors to employ what I consider to be probably the most appropriate use ever of random forest regression in a microbial ecology study (even though it only worked until the main source of the microorganisms switched, which perhaps made it unexpectedly damning evidence that this approach has limited utility ultimately). The results are predictable but nevertheless very interesting from the perspective of priority effects (or the apparent lack thereof). I think the manuscript is absolutely acceptable for publication after some very minor improvements, and I encourage the authors to consider a couple of points that they hadn't explicitly mentioned.

Specific comments:

- Before microorganisms associated with rain washed everything else out at the one-year mark, the apparent lack of priority effects (i.e., "predicable [sic] nature of microbial colonization"), which the authors attributed to the harsh environments, contradicts the complex and spatially heterogeneous microbial biogeography observed in many arid (i.e., harsh) habitats. Perhaps this is a function of time, and priority effects manifest over much longer timeframes than covered in this study, or perhaps newly cooled lava imposed such strong selective forces that there is only one straight and narrow path for primary succession. Another question is why the 'rain microbiome' apparently arrested the succession processes, at least within the timeframe of the study. At what point does new lava begin to resemble old lava microbiologically, and what factors may trigger that shift? I would like to see the authors' thoughts on these points.

We agree that priority effects result in self-organizational patterns and spatial heterogeneity in many extreme environments, including volcanic systems. For example, Selensky et al. (2021) showed that despite organic matter inputs, biofilms within lava tubes at Lava Beds National Monument were supported primarily by active C fixation. Chemoautotrophs could have been the initial colonizing group in the newly formed lava tube and established a unique niche by providing fixed C to heterotrophs. As overlying vegetation was established and organic detritus inputs increased, the original chemoautotroph colonizers may have maintained their independent niche, illustrating the importance of priority effects in volcanic systems. In our work, these differences may have been too subtle to detect over the limited timescales due to the similar environmental conditions across sites and eruptions (e.g., all eruptions took place in the summer from the same volcanic system with similar chemistry). Accordingly, the consistent microbial assembly patterns, presence of homogenous selection, and generally high NRI/NTI values suggest strong environmental filtering early on. Nonetheless, drift was also determined to be present along with lower NRI values in some samples, suggesting that stochastic events also had an influence on successional processes. There were subtle differences between sites that would likely compound

over longer timescales. For example, as we say in the text (Lines 420–422): “LME models applied to a variety of metrics found that the site location accounted for some of the variance in the data (e.g., 40.8% for Faith’s PD; Supplementary Table 5), underscoring some influence of local environmental conditions.” Therefore, it is plausible that early microbial colonization is governed primarily by deterministic filters, while priority effects may emerge later as species establish more permanent niches and micro-environmental heterogeneity compounds. Additionally, as lichen biocrusts, moss, and other flora start to colonize the lava, the microbiology might start to increasingly resemble the old lava, but these processes would occur on the timescales of decades to centuries rather than a few years. We are currently working on follow-up analyses that explore these questions on how micro-environmental conditions impact biomass and biodiversity distribution in volcanic systems at different timescales. Meanwhile, to address these points here, we have adjusted the paragraph in the discussion on the random forest model (Pages 21–22, Lines 449–465):

“While early colonization is likely governed primarily by deterministic processes, the influence of priority effects, biotic interactions, and micro-environmental heterogeneity may compound over longer timescales, potentially limiting the generalizability of machine-learning models trained on early-stage data. For example, Selensky et al.⁶⁵ found that despite organic matter inputs, biofilms within lava tubes at Lava Beds National Monument were supported primarily by active carbon fixation. Chemoautotrophs likely were the initial colonizers in the newly formed lava tube and as overlying vegetation was established and organic detritus inputs increased, the original chemoautotroph colonizers may have maintained their independent niche. Similarly, we show that NRI decreased over time and there was a shift towards stochastic assembly (Fig. 6), reflecting the increasing importance of biotic factors (e.g., species interactions) and likely the early stages of compounding priority effects.”

We also edited Page 19, Lines 400–408 to further highlight the ecological importance of the shift towards rain-sourced microorganisms:

“Source tracking (Fig. 5) revealed that rainwater became the predominant contributor to microbial assemblages on the lava flows in all three eruptions after the first winter, suggesting that local precipitation events increasingly shaped community composition. This transition potentially reflects a broader ecological shift toward more stable microbial establishment and emergence of biotic interactions. Moreover, this demonstrates that rain may be an important driver for microbial dispersal around the world.”

- Why did the authors think that a regression based on PAPRICA outputs would increase the generalizability of the model? 'Predicted' metagenomes from PAPRICA, PICRUST, or Tax4Fun by nature are based on correlations and cannot introduce new independent variables.

See our response regarding these tools and the adjustments made on page 1 of this document. Briefly, we added additional tools for robustness and discussed caveats to ensure that the reader understands that the metabolic inference pipelines should be

interpreted with caution. We include these results as an exploratory analysis to see if there are possible metabolic shifts that can be predictive of the successional stage.

- Please refer to relevant figures and tables in the Results section consistently (e.g., line 187).

We mistakenly left out a taxa bar plot figure for the source environments and fumaroles in the Supplementary. We have added this figure (as Supplemental Fig. 3) and added references to the figure where appropriate throughout the text.

- The authors should do a spellcheck and a careful proofread. See [sic] above.

We have conducted a thorough spellcheck and proofread.

- Please italicize all Latin taxonomic names as per <https://journals.asm.org/writing-your-paper><<https://url.usb.m.mimecastprotect.com/s/C1iWC3YmkgfRVyWK3lruliQgpF-?domain=journals.asm.org>>#nomenclature

Taxonomic names from phyla level down are now italicized.

- Some of the symbols in Figure 1a are too small, and it is unclear how many there are for each type. Figure 1b is quite difficult to read--consider making the labels bright yellow.

We have increased the size of markers in both subplots to improve readability and made the font color brighter in the labels in Figure 1b. Additionally, we added the following to the caption on Figure 1:

“A total of 18 Fagradalsfjall, 3 Meradalir, 4 Litli-Hrútur sites were established, though some are too closely spaced to be individually resolved on the map.”

Reviewer #3 (Remarks to the Author):

This manuscript presents a comprehensive and well-executed examination of microbial colonization on newly emplaced lava flows in Iceland. The authors leverage the 2021-2023 Fagradalsfjall eruptions as a natural experiment to document the transition from sterile substrate to inhabited ecosystem with unprecedented temporal resolution - from hours to years post-emplacement. The multi-faceted approach combining 16S rRNA sequencing, cell counts, culturing, and statistical modeling provides robust evidence for a predictable two-stage colonization process.

The study's major strengths lie in its exceptional temporal resolution, the fortuitous natural replication provided by three sequential eruptions, and the integration of multiple data streams to build a comprehensive picture of community assembly. The random forest model's ability to predict successional stage across eruptions with high accuracy ($R^2 = 0.95$) is particularly compelling and demonstrates the predictable nature of these assembly processes.

While the authors have thoughtfully addressed spatial variability by treating site as a random effect in their linear mixed effects models, I wonder if there might be systematic differences in environmental parameters between sampling locations that could influence colonization patterns beyond what's captured. The authors note that pH and water content remained relatively constant, but could there be gradients in lava chemistry across flows or systematic differences in exposure to surrounding microbiomes? The spatial distance from old to new lava spans potentially distinct soil/water/air microbiomes that could create a bias in what's emigrating to the new surfaces. Similarly, while all lava flows presumably start chemically similar, subtle differences between eruption events could create environmental biases that compound with temporal effects.

Even within a single volcanic system, subtle differences in lava chemistry can influence microbial community assembly. For example, previous work in seafloor volcanic systems has demonstrated that distinct mineral substrates can host different microbial communities despite close geographic proximity (e.g., Toner et al., 2012). In our study, however, the mineralogy across the three eruptions is broadly similar, as shown in our XRD data (Supplementary Table 1). All flows were derived from similar peridotitic mantle sources, and major geochemical differences were largely restricted to the early phase of the 2021 eruption, which exhibited high compositional variability. In addition to composition, lava morphology, which dictates surface roughness, porosity, and microenvironmental conditions may also drive microbial colonization dynamics. As already noted on Page 5, Lines 91–94, the 2022 Meradalir eruption produced friable, porous spatter on a slope, and cell counts there after one year were comparable to those at the Fagradalsfjall eruption after two years. Potentially, mass wasting (e.g., soil deposition on the lava) and higher porosity contributed to higher cell counts. Similarly, Sites 1–4 were also emplaced against a slope which would explain the higher proportion of soil particles in the SourceTracker2 results (already discussed on Page 15, Lines 324–328). We are currently conducting a follow-up study that investigates these patterns more explicitly, linking spatial heterogeneity in lava structure and local environmental context to microbial biomass and diversity. To acknowledge these nuances here, however, we have added/modified the following text to the Discussion (Page 20, Lines 421–431):

“LME models applied to a variety of metrics found that the site location accounted for some of the variance in the data (e.g., 40.8% for Faith's PD; Supplementary Table 5), underscoring some influence of local environmental conditions. Some of this site-to-site variability may reflect differences in lava morphology and microenvironmental

exposure⁴. For instance, porous spatter from the Meradalir eruption appeared to support more rapid colonization than Fagradalsfjall samples, with elevated cell counts after one year, potentially due to increased surface area, friability, or slope-driven soil input (Supplementary Fig. 1A). Additionally, while all three eruptions potentially originated from similar mantle regions⁶² and our samples exhibited broadly similar mineralogy (Supplementary Table 1), subtle geochemical differences—possibly reflecting variable source contributions^{24,63}—could have had compounding effects in microbial community assembly⁶⁴.”

The disparity between community structure and metabolic function predictability ($R^2 = 0.95$ vs 0.64) is expected but intriguing. As the authors acknowledge, this likely stems from both sparse metabolic annotation data and the physiological plasticity/functional redundancy inherent in microbial communities. This gap between "who's there" and "what they're doing" remains one of the field's persistent challenges.

As discussed on page 1 of this document, this has been modified to emphasize that these metabolic inference pipelines are meant to be exploratory and should be interpreted with caution. We also include additional pipelines for robustness.

What I find most compelling is the observed decrease in NRI over time, suggesting a fundamental shift in assembly drivers. The initial harsh conditions of fresh lava create strong abiotic filters - only the hardiest arrive and survive. But as time progresses and community coalescence increases, biotic interactions increasingly shape assembly. The emigrating microbes from surrounding environments must now contend with an established lava surface community, making competition, resource partitioning, and other biotic factors increasingly important. This reminds me of priority effects in classical ecology - who gets there first matters.

We added additional material to the discussion (Page 21, Lines 447–458) regarding priority effects, and discuss this in detail on pages 7 and 8 of this document. Briefly, it is possible that environmental filtering was the primary control on community assembly in the early stages, but priority effects and biotic interactions compound over time and we just were beginning to observe their effects on the timescales of this study (e.g., as evidenced by decreasing NRI).

This pattern contrasts interestingly with other primary succession studies. Nemergut et al.'s (2016) beach microcosm work found decreases in rRNA operon copy number during succession, inferring a shift from r-selected early colonizers to K-selected later arrivals. But their system was nutrient-rich - quite the opposite of these harsh lava flows. I wonder if the authors' data might reveal an inverted pattern? Perhaps the extreme conditions initially favor K-selected organisms capable of surviving stress and efficiently using limited resources, rather than fast-growing opportunists. Examining rRNA copy number patterns could provide fascinating insights into colonization strategies.

As discussed on page 1 of this document, using metabolic inference tools (e.g., PICRUST2 used by Nemergut et al.) must be treated with extreme caution. However,

since we have already added PICRUSt2 to our random forest model, we have added a plot to the supplementary file illustrating the weighted mean copy number of each sample as a function of time. Similar to Nemergut et al. (2016), which surveyed several different types of environments, we also find a decrease in copy number. To address these results, we have included the following text to the results on Page 15, Lines 312–316:

“As an additional example, PICRUSt2-estimated 16S rRNA gene copy number decreased over time across all eruptions (LME model, $p < 0.001$; Supplementary Fig. 8), indicating a shift toward slower-growing microorganisms, consistent with patterns observed in successional studies such as Nemergut et al.⁵⁶. However, the overall copy numbers remained comparatively low.”

The increasing influence of community coalescence with time deserves deeper exploration. As succession progresses, the introduction of new species from neighboring areas coupled with an established community should create a complex mosaic of interactions. Rather than a simple, predictable successional trajectory, we might expect increasing spatial and temporal heterogeneity as these mixing processes intensify. Yet remarkably, the authors still find predictable patterns - perhaps testament to the overriding influence of the harsh lava environment even as coalescence increases.

Despite the potential for increased spatial and temporal heterogeneity due to microenvironmental variations and priority effects, we observed relatively consistent successional trajectories across eruptions and sites, which likely reflect the strong deterministic filtering imposed by the lava (Fig. 6). However, as the system began to stabilize after the first winter, there was a shift towards dispersal limitation and lower NRI/NTI, demonstrating the increased influence of stochasticity. Despite this shift, the

microbial community composition continued to change in a consistent, directional manner with time across all eruptions. This suggests that while we may have been beginning to observe divergence in community trajectories, such changes have not yet overridden the strong temporal signal during the 3-year timeframe examined here. Byloos et al. (2018) showed that environmental differences were more significant than age differences in microbial communities in ~decades old lavas in Iceland. In more temperate or nutrient-rich environments, these shifts toward stochasticity and heterogeneity may occur earlier and more rapidly. We have added/modified the following text to the Discussion near the discussion of priority effects to address these points (Page 21–22, Lines 457–465):

“Similarly, we show that NRI decreased over time and there was a shift towards stochastic assembly (Fig. 6), reflecting the increasing importance of biotic factors and likely the early stages of compounding priority effects. In decades-old lava, previous work found that differences of a few years did not significantly impact the microbial community, likely due to environmental differences¹⁷. These findings suggest that spatial heterogeneity, stochastic processes, and priority effects may increase in influence over time—but have not yet overridden the strong temporal signal observed within the three-year window examined here (Fig. 7).”

A few additional considerations: The ephemeral fumarole communities remaining arrested in an early successional state despite being one year old is particularly intriguing. This suggests that even modest environmental stress can override temporal progression, keeping communities in a perpetual pioneer state. Also, the shift from aerosol/soil sources to rain-derived communities after the first winter represents a fundamental change in colonization dynamics that warrants further investigation.

We discuss in our recent review article (Hadland et al., 2024) the differences between fumarole- and lava-hosted microbial communities in detail. The dynamics of community assembly may depend on the local environment, since in extreme systems, fumaroles may provide abodes of habitability while in temperate environments, the surrounding lava may impose fewer selection pressures comparatively. This supports the idea that harsh microsite conditions can delay or arrest succession, resulting in spatial heterogeneity. We already discuss most of these points on Page 18–19, Lines 386–399, but have added the following text to the end of the paragraph (Lines 396–399) to emphasize the point further:

“Consequently, localized environmental stress might have suppressed succession, and the elevated temperatures may have restricted colonization to a narrow set of stress-tolerant taxa with limited niche width, slowing community turnover.”

Finally, we expanded our discussion of rain-derived microorganisms in response to a comment by R2 on page 8 of this document.

In summary, this work makes significant contributions to our understanding of microbial primary succession in extreme environments. The documentation of the transition from

abiotic to biotic assembly drivers, replicated across multiple eruptions, provides rare empirical support for theoretical predictions about community assembly. The predictive modeling demonstrates that microbial communities can serve as reliable indicators of ecosystem development stage. With some expanded discussion of environmental covariates, functional predictions, and the complex interplay between harsh conditions and community coalescence, this manuscript will be of broad interest to microbial ecologists and those studying ecosystem development under extreme conditions.